# Dopamine neuron dependent behaviors mediated by glutamate cotransmission

**Susana Mingote[1,2]\*, Nao Chuhma[1,2], Abigail Kalmbach[1,2], Gretchen M Thomsen[1], Yvonne Wang[1], Andra Mihali[1], Caroline Sferrazza[1], Ilana Zucker-Scharff[1], Anna-Claire Siena[2], Martha G Welch[1,3,4], José Lizardi-Ortiz[5], David Sulzer[1,2,5,6], Holly Moore[1,7], Inna Gaisler-Salomon[1,8], Stephen Rayport[1,2]\***

[1]Department of Psychiatry, Columbia University, New York, United States; [2]Department of Molecular Therapeutics, NYS Psychiatric Institute, New York, United States; [3]Department of Pediatrics, Columbia University, New York, United States; [4]Department of Developmental Neuroscience, NYS Psychiatric Institute, New York, United States; [5]Department of Neurology, Columbia University, New York, United States; [6]Department of Pharmacology, Columbia University, New York, United States; [7]Department of Integrative Neuroscience, NYS Psychiatric Institute, New York, United States; [8]Department of Psychology, University of Haifa, Haifa, Israel

**Abstract** Dopamine neurons in the ventral tegmental area use glutamate as a cotransmitter. To elucidate the behavioral role of the cotransmission, we targeted the glutamate-recycling enzyme glutaminase (gene *Gls1*). In mice with a dopamine transporter (*Slc6a3*)-driven conditional heterozygous (cHET) reduction of *Gls1* in their dopamine neurons, dopamine neuron survival and transmission were unaffected, while glutamate cotransmission at phasic firing frequencies was reduced, enabling a selective focus on the cotransmission. The mice showed normal emotional and motor behaviors, and an unaffected response to acute amphetamine. Strikingly, amphetamine sensitization was reduced and latent inhibition potentiated. These behavioral effects, also seen in global GLS1 HETs with a schizophrenia resilience phenotype, were not seen in mice with an *Emx1*-driven forebrain reduction affecting most brain glutamatergic neurons. Thus, a reduction in dopamine neuron glutamate cotransmission appears to mediate significant components of the GLS1 HET schizophrenia resilience phenotype, and glutamate cotransmission appears to be important in attribution of motivational salience.

\*For correspondence: sm2964@cumc.columbia.edu (SM); sgr1@columbia.edu (SR)

**Competing interests:** The authors declare that no competing interests exist.

## Introduction

Dopamine (DA) neurons regulate several aspects of motivated behaviors (*Bromberg-Martin et al., 2010*; *Salamone and Correa, 2012*; *Schultz, 2013*), and are involved in the pathophysiology of neuropsychiatric disorders ranging from drug dependence to schizophrenia (*Robinson and Berridge, 2008*; *Winton-Brown et al., 2014*). Like most CNS neurons, DA neurons release multiple neurotransmitters (*Trudeau et al., 2014*). They release DA with both slower modulatory actions (*Tritsch and Sabatini, 2012*), as well as faster signaling actions (*Ford et al., 2009*; *Chuhma et al., 2014*). They variously release glutamate (GLU) (*Hnasko and Edwards, 2012*) and GABA (*Tritsch et al., 2016*) as cotransmitters, conferring both greater dynamic signaling range and heterogeneity in their synaptic actions, as well as differential susceptibility to endogenous and exogenous modulation (*Chuhma et al., 2017*). Discerning the behavioral role of DA neuron GLU cotransmission has been challenging (*Morales and Margolis, 2017*).

**eLife digest** A small cluster of neurons found in the midbrain use dopamine to send signals to neurons involved in many processes including motivation and attention. Drugs of abuse such as amphetamine co-opt motivation by increasing dopamine signaling. When used excessively, the drugs can engender delusional thinking, as is seen in schizophrenia. In contrast, the drugs used to treat schizophrenia block excess dopamine signaling. Recently it has been shown that dopamine neurons in the middle part of the midbrain release both dopamine and glutamate. The exact role of this dopamine neuron glutamate signaling has been difficult to find out.

Previous experiments involved genetically modifying dopamine neurons so that they would not release glutamate. However, this affected how the neurons develop, making it difficult to discern the effects of glutamate signaling. Now, in genetically modified mice that have less glutaminase in their dopamine neurons than normal, Mingote et al. find that glutamate signaling is reduced just when dopamine neurons fire more rapidly. This did not change how dopamine neurons develop or how they use dopamine to signal.

This reduction in dopamine neuron glutamate signaling affects two behaviors that are driven by the activity of dopamine neurons. First, it reduces the effects of a process called amphetamine sensitization, in which repeated doses of amphetamine increase dopamine neuron signaling so that events associated with drug use take up more attention than they normally would. Second, the modified mice were better able to ignore familiar, irrelevant sounds in their environment; the mice continued to pay less attention to a familiar sound, even when it was paired with a shock and came to predict an unpleasant event – a process known as potentiation of latent inhibition. The effects on both of these processes suggest that dopamine neuron glutamate signaling helps animals decide which features of their environment are most important.

This result suggests a new way of treating schizophrenia. When humans take amphetamine repeatedly, which produces sensitization, they can develop psychosis, a principal symptom of schizophrenia. During a period of psychosis, thoughts and perceptions are disturbed, making it difficult to distinguish between relevant or irrelevant things in the environment. By reducing amphetamine sensitization and potentiating latent inhibition, blocking dopamine neuron glutamate signaling might help to treat the symptoms of schizophrenia.

DA neuron GLU cotransmission has a crucial neurodevelopmental role. The abrogation of GLU cotransmission via a DA transporter (DAT)-driven conditional knockout (cKO) of vesicular GLU transporter 2 (VGLUT2, encoded by *Slc17a6*) (*Hnasko et al., 2010*; *Stuber et al., 2010*) impairs survival and axonal arborization of DA neurons in vitro, and compromises the development of the mesostriatal DA system in vivo leading to a 20% decrease in the number of DA neurons (*Fortin et al., 2012*). GLU cotransmission also plays an important role in modulating DA release by enhancing packing of DA into vesicles (*Hnasko et al., 2010*) via vesicular synergy (*El Mestikawy et al., 2011*). Functionally, DAT VGLUT2 cKO show about a 25% reduction in electrically-evoked DA release and about a 35% reduction in DA content in the nucleus accumbens (NAc) (*Hnasko et al., 2010*; *Fortin et al., 2012*). Behaviorally, DAT VGLUT2 cKO show modest deficits in emotional and motor behaviors (*Birgner et al., 2010*; *Fortin et al., 2012*), normal reinforcement learning drive by DA neuron activation but decreased response vigor (*Wang et al., 2017*), a blunted response to psychostimulants (*Birgner et al., 2010*; *Hnasko et al., 2010*), and a paradoxical increase in sucrose and cocaine seeking (*Alsiö et al., 2011*). Whether the behavioral phenotypes of DAT VGLUT2 cKO mice are due to the impact of the VGLUT2 deficit on DA neuron development, DA transmission, or GLU synaptic actions is not clear.

Phasic activity of DA neurons projecting to the NAc encodes the incentive salience of reward-predicting cues and invigorates cue-induced motivated behaviors (*Bromberg-Martin et al., 2010*; *Flagel et al., 2011*). At the synaptic level in the striatum, DA neurons make the strongest GLU connections in the NAc shell to cholinergic interneurons (ChIs) (*Chuhma et al., 2014*; *Mingote et al., 2015*). When DA neurons are driven at burst firing frequencies — mimicking their in vivo phasic firing

— their GLU postsynaptic actions drive synchronized burst-pause sequences in ChIs (*Chuhma et al., 2014*) that are likely to be important in salience encoding.

Dysregulated DA neuron firing is thought to disrupt salience processing leading to the development of psychotic symptoms (*Kapur, 2003*; *Winton-Brown et al., 2014*). The hyperdopaminergic state associated with positive symptoms of schizophrenia is modeled in rodents by amphetamine sensitization (*Peleg-Raibstein et al., 2008*), which enhances the motivational salience of drug-associated stimuli (*Robinson et al., 2016*). Interestingly, amphetamine sensitization as well as gestational MAM treatment, a validated rodent model of schizophrenia, selectively enhance activity of VTA neurons projecting to NAc shell (*Lodge and Grace, 2012*), a key brain region associated with motivational salience (*Ikemoto, 2007*), where DA neurons make the strongest GLU connections (*Mingote et al., 2015*). Dysregulation in salience processing is also thought to underlie the disruption of latent inhibition (LI) seen in schizophrenia (*Weiner, 2003*). Disruption of LI is replicated in rodents by amphetamine-induced increases in DA neuron activity (*Young et al., 2005*), in particular increases in DA neuron phasic firing (*Covey et al., 2016*). Although DA neuron GLU signals at burst frequencies control NAc shell activity, it remains to be established whether GLU cotransmission is necessary for the expression of behaviors dependent on salience attribution and associated with schizophrenia.

So we sought to temper GLU release at the higher firing frequency of bursts, independent of DA release. For this we targeted phosphate-activated glutaminase (PAG), encoded by *Gls1,* in order to reduce presynaptic glutamate synthesis modestly without affecting DA neuron vesicular dynamics, as well as minimizing effects on DA neuron development. Most presynaptic GLU arises from the action of PAG; once released, GLU is taken up by neighboring astrocytes, metabolized to glutamine, and transferred back to presynaptic terminals where it is converted to GLU by PAG (*Marx et al., 2015*). This GLU–glutamine cycle is particularly important in sustaining GLU release with higher frequency firing (*Billups et al., 2013*; *Tani et al., 2014*). Indeed, deletion (*Masson et al., 2006*) or heterozygous reduction of *Gls1* (*Gaisler-Salomon et al., 2009b*) decreases GLU neurotransmission at higher firing frequencies selectively. The global heterozygous *Gls1* reduction impacts several DA dependent behaviors that underpin a schizophrenia resilience phenotype (*Gaisler-Salomon et al., 2009b*), characterized by an attenuated response to psychostimulant challenge, potentiated latent inhibition, procognitive effects (*Hazan and Gaisler-Salomon, 2014*), together with CA1 hippocampal hypoactivity inverse to the CA1 hyperactivity seen in patients with schizophrenia (*Gaisler-Salomon et al., 2009a*; *Schobel et al., 2009*). Genetic mutations engendering resilience carry strong therapeutic valence as they directly identify therapeutic targets (*Mihali et al., 2012*).

Here we show in DAT GLS1 conditional heterozygous (cHET) mice — with a DAT (*Slc6a3*)-driven *Gls1* reduction — that DA neuron GLU cotransmission is reduced in a frequency dependent manner, without affecting DA neuron development or DA release, and that behaviors that rely on the motivational salience-encoding function of DA neurons are selectively affected, with implications of DA neuron GLU cotransmission for schizophrenia pharmacotherapy.

## Results

### Expression of PAG in DA neurons

DA neurons immunoreactive for PAG are found in both the ventral tegmental area (VTA) and substantia nigra pars compacta (SNc) in rat (*Kaneko et al., 1990*), but this has not been examined in mouse. Moreover, the expression of PAG in DA neurons has never been addressed stereologically. We immunostained ventral midbrain sections of the VTA and SNc for the DA-synthetic enzyme tyrosine hydroxylase (TH) and for PAG (*Figure 1A*; *Figure 1—figure supplement 1A*). This revealed TH positive ($^+$) / PAG$^+$, PAG only (TH negative ($^-$) / PAG$^+$), or TH only (TH$^+$ / PAG$^-$) neurons (*Figure 1B*). Stereological counts in P25 mice showed that the three cell populations were present in similar proportions in the VTA and SNc (*Figure 1C*). In contrast, DA neurons expressing VGLUT2 are concentrated in the medial VTA (*Yamaguchi et al., 2015*).

Since DA neurons capable of GLU cotransmission express VGLUT2 (*Hnasko et al., 2010*; *Stuber et al., 2010*) and the majority of neurotransmitter GLU is produced by PAG, DA neurons expressing VGLUT2 should preferentially express PAG. To determine the number of DA neurons expressing both VGLUT2 and PAG mRNA, we performed a single cell reverse transcription (RT)-PCR

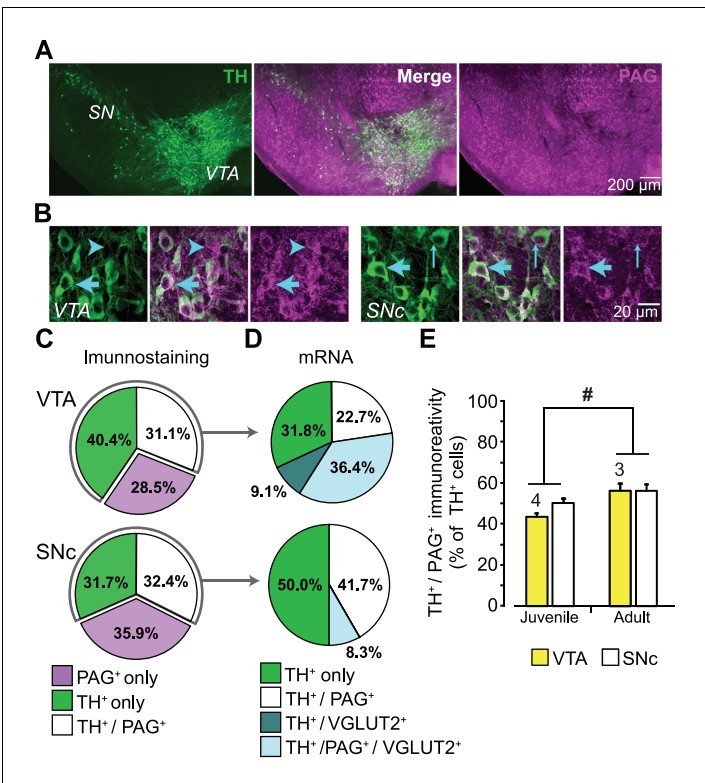

**Figure 1.** Expression of phosphate-activated glutaminase (PAG) in mouse ventral midbrain DA neurons. (**A**) Confocal mosaic z-projected image of the ventral midbrain showing TH (green, left) and PAG (magenta, right) immunoreactivity. Merged image (center) shows that some TH$^+$ DA neurons co-express PAG (white). The specificity of the PAG antibody was verified in GLS1 KO mice; see *Figure 1—figure supplement 1A*. (**B**) Magnified confocal images in the VTA (left) and SNc (right) showing TH$^+$ only (thin blue arrow), PAG$^+$ only (blue arrow head) and TH$^+$/PAG$^+$ cells (thick blue arrow). (**C**) Stereological counts of TH$^+$ only (green), PAG$^+$ only (magenta) and TH$^+$ / PAG$^+$ (white) cells in the VTA and SNc of juvenile (P25) wild type mice (n = 4). Cell numbers in the VTA (TH$^+$only = 4681, PAG$^+$only = 3411, TH$^+$ / PAG$^+$=3673) were greater than in the SNc (TH$^+$only = 2564, PAG$^+$only = 2909, TH$^+$ / PAG$^+$=2595) (two-way ANOVA: main effect of brain region, $F_{(1,18)}$= 18.36; p<0.001; effect size (ES) partial $\eta^2$ = 0.51), but the relative proportions of cell types did not differ between regions(main effect of cell type, $F_{(2,18)}$= 1.22; p=0.318; cell type X brain region interaction, $F_{(2,18)}$= 2.70; p=0.094). (**D**) Single-cell RT-PCR analysis of cells expressing TH mRNA, in the VTA and SNc of juvenile mice (P25-37), showing the percentage of cells that co-expressed PAG and VGLUT2 mRNA. In the VTA, most cells were either TH$^+$ only (7/22) or TH$^+$/PAG $^+$/ VGLUT2$^+$(8/22); there were also TH$^+$/PAG$^+$ cells (5/22) and rarely TH$^+$/VGLUT2$^+$ (2/22). In the SNc, most cells were either TH$^+$ only (5/12) or TH$^+$/PAG$^+$ cells (6/12); and rarely TH$^+$/PAG$^+$/VGLUT2$^+$ (1/12). No TH$^+$ cells expressed GAD mRNA. For the full coexpression analysis, including GAD mRNA, see *Figure 1—figure supplement 1B and C*. (**E**) Comparison of the relative number of TH$^+$ / PAG$^+$ cells in juvenile (P25) and adult (P60) mice. In both the VTA and SNc, there was a significant increase in the number of TH$^+$ / PAG$^+$ cells. # indicates a significant main effect of age (two-way ANOVA, $F_{(1,10)}$= 8.26; p=0.017, ES partial $\eta^2$ = 0.45); there was no significant region effect ($F_{(1,10)}$= 2.154; p=0.173), nor interaction, ($F_{(1,10)}$= 0.846; p=0.379). See *Figure 1—source data 1*.xlsx for source data and all statistical analysis.

The following source data and figure supplement are available for figure 1:

**Source data 1.** Stereology of TH and PAG positive cells in the VTA and SNc in Juvenile and Adult.
**Figure supplement 1.** Expression of PAG in dopamine neurons.

analysis in P25-37 mice (*Figure 1D*; *Figure 1—figure supplement 1B*). Since DA neurons also corelease GABA (*Tritsch et al., 2016*), which could derive in part from glutamic acid decarboxylase (GAD) metabolism of GLU (produced by PAG), we also examined the expression of GAD67 mRNA (*Figure 1—figure supplement 1B*). We found that VGLUT2 mRNA was highly concentrated in VTA DA neurons but rarely expressed in SNc DA neurons. Importantly, $TH^+$ / $VGLUT2^+$ neurons preferentially expressed PAG (9 out of 11 $TH^+$ / $VGLUT2^+$ cells coexpressed PAG; $\chi^2$ =3.6, p=0.035), further supporting the role of PAG in GLU cotransmission (*Figure 1D*). GAD67 was not found in $TH^+$ / $PAG^+$ neurons; while a few DA neurons expressed GAD67 (*Kim et al., 2015* supplemental information), a larger sample would be required to assess the role of PAG in GABA cotransmission. Yet, some $TH^-$ / $PAG^+$ neurons in both the VTA (2/6 cells) and SN (6/13 cells) were $GAD^+$, identifying them as GABA neurons and suggesting that PAG contributes to GABA synthesis in those neurons (*Figure 1—figure supplement 1B,C*). We also found $TH^-$ / $PAG^+$ VTA neurons that coexpress VGLUT2 (*Figure 1—figure supplement 1B,C*), identifying them as GLU neurons (*Hnasko et al., 2012*; *Yamaguchi et al., 2015*). Given that coexpression of VGLUT2 decreases with maturation (*Trudeau et al., 2014*), we compared the number of $TH^+$ / $PAG^+$ neurons in juvenile (P25) and adult (P60) wild-type mice. The number of $PAG^+$ / $TH^+$ neurons in both the VTA and SNc increased modestly with age (*Figure 1E*). Although DA neurons throughout the ventral midbrain express PAG, only medial DA neurons that also express VGLUT2 are capable of GLU cotransmission, so the impact of a PAG reduction on GLU cotransmission should be further restricted to VGLUT2-expressing DA neurons.

## Conditional *Gls1* reduction in DA neurons

To address the specific function of DA neuron GLU cotransmission, we bred DAT^IREScre mice (*Bäckman et al., 2006*) with *floxGls1* mice (*Mingote et al., 2016*) to reduce *Gls1* coexpression selectively (*Figure 2*). DAT and other DA neuron specific gene expression is not affected in the ventral midbrain and striatum of DAT^IREScre HET mice (*Bäckman et al., 2006*), which we confirmed (*Figure 2—figure supplement 1A*); the acute locomotor response to amphetamine, a drug that targets

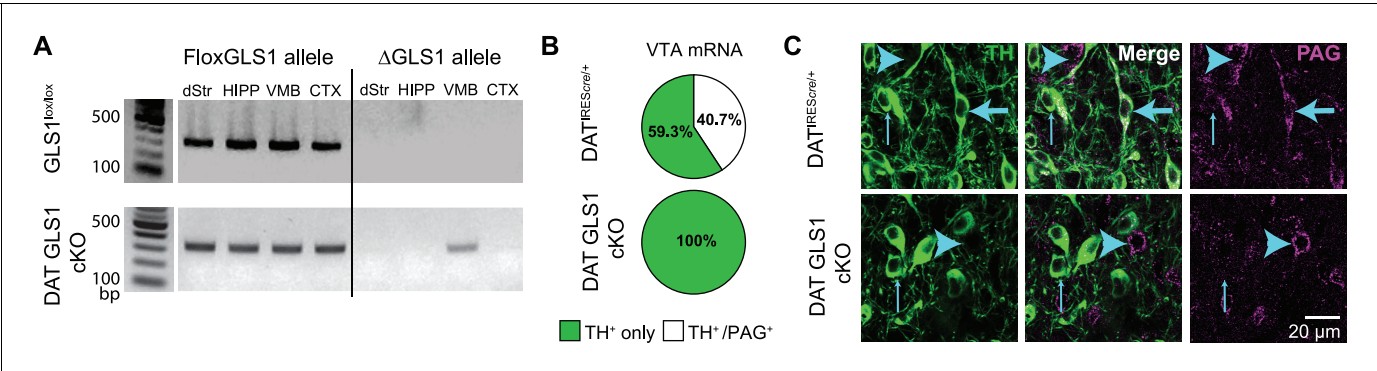

**Figure 2.** DA neuron selective PAG deletion. (**A**) PCR screens for the floxGLS1 allele (left) and ΔGLS1 allele (right) in brain regions from both GLS1^lox/lox and DAT GLS cKO mice. The ΔGLS1 allele was present solely in DAT GLS1 cKO ventral midbrain. dStr, dorsal striatum; HIPP, hippocampus; VMB, ventral midbrain; CTX, cortex. Gel is representative of 3 replications. (**B**) Single-cell rtPCR analysis of TH expressing cells in the VTA in DAT^IREScre/+ and DAT GLS1 cKO mice. In the VTA of DAT^IREScre/+ mice, 11/30 TH cells expressed PAG mRNA, while in DAT GLS1 cKO none did (0/38 cells). (**C**) Confocal photomicrographs of the VTA from DAT^IREScre/+ and DAT GLS1 cKO mice showing $TH^+$ only (thin blue arrow) and $PAG^+$ only (blue arrow head) and $TH^+$/$PAG^+$ cells (thick blue arrow). There were no $TH^+$/$PAG^+$ cells in the DAT GLS1 cKO ventral midbrain. Expression of dopaminergic markers and amphetamine-induced hyperlocomotion were not affected in DAT^IREScre mice; see *Figure 2—figure supplement 1*. These mice were control (CTRL) mice in subsequent experiments.

The following source data and figure supplements are available for figure 2:

**Figure supplement 1.** Expression of dopaminergic markers and amphetamine-induced hyperlocomotion were not affected in DAT^IREScre mice.

**Figure supplement 1—sourcedata 1.** Expression of dopaminergic markers and amphetamine-induced hyperlocomotion in DAT^IREScre mice .

DAT function, was also not affected (*Figure 2—figure supplement 1B,C*). We have shown previously that *Gls1* expression from the *floxGls1* allele is normal (*Mingote et al., 2016*).

Conditional targeting was verified in DAT GLS1 cKO mice (DAT$^{IREScre/+}$::GLS1$^{lox/lox}$) mice. PCR screens of genomic DNA showed the non-functional truncated ($\Delta$) *Gls1* allele in the ventral midbrain of DAT GLS1 cKO mice, but not in forebrain regions that do not contain DA neurons, the dorsal striatum (dStr), frontal cortex and hippocampus (*Figure 2A*). We used single cell RT-PCR analysis to verify further the *Gls1* inactivation in DA neurons (*Figure 2B*). In DAT GLS1 cKO mice, *Gls1* mRNA was absent in VTA cells expressing TH mRNA. There was no impact on the number of DA neurons that expressed VGLUT2 (3/38 in DAT$^{IREScre/+}$ vs. 6/30 DAT GLS1 cKO mice, $\chi^2$ =1.2, p=0.27). To confirm the conditional strategy at the protein level, we examined TH and PAG immunoreactivity in the VTA (*Figure 2C*). In DAT GLS1 cKO mice, all TH$^+$ cells were PAG$^-$, while neighboring TH$^-$ but PAG$^+$ cells were seen, demonstrating the specificity of *Gls1* targeting. Since heterozygous reduction in *Gls1* is sufficient to attenuate GLU transmission at higher-firing frequencies (*Gaisler-Salomon et al., 2009b*), and to minimize compensatory mechanisms seen in KOs (*Bae et al., 2013*), we used DAT GLS1 cHET mice (DAT$^{IREScre/+}$::GLS1$^{lox/+}$) and DAT$^{IREScre/+}$ mice as controls (CTRL).

## Frequency-dependent attenuation of GLU cotransmission in DAT GLS1 cHETs

To measure DA neuron synaptic transmission, we conditionally expressed channelrhodopsin 2 (ChR2) in DA neurons using Ai32 (RCL-ChR2(H134R)/EYFP) mice (*Madisen et al., 2012*), to obtain triple mutant DAT GLS1 cHET::ChR2 (DAT$^{IREScre/+}$::GLS1$^{lox/+}$::Ai32) and double mutant control CTRL::ChR2 (DAT$^{IREScre/+}$::Ai32) littermates. We confirmed that the expression of ChR2-EYFP was specific to DA neurons independent of *Gls1* genotype (*Figure 3—figure supplement 1A,B*). We also confirmed that TH$^+$ / DAT$^-$ striatal interneurons (*Xenias et al., 2015*) do not express ChR2-EYFP (*Figure 3—figure supplement 1C*). We then examined the impact of PAG deficiency on GLU cotransmission in recordings from cholinergic interneurons (ChIs) and spiny projection neurons (SPNs) in the NAc medial shell, the striatal hotspot for DA neuron GLU transmission (*Chuhma et al., 2014*; *Mingote et al., 2015*) (*Figure 3A*). ChIs and SPNs were identified by soma size and electrophysiological signature, under current clamp (*Figure 3—figure supplement 2A*). We confirmed that the intrinsic membrane properties of ChIs and SPNs did not differ between genotypes (*Figure 3—figure supplement 2B,C,D,E*).

We measured DA neuron GLU cotransmission in DAT GLS1 cHET::ChR2 mice (P60-P76) in the NAc shell with single pulse photostimulation (5 ms duration, delivered with a 10 s interval) and burst photostimulation (5 pulses at 20 Hz, delivered with a 30 s interval) of DA neuron terminals. The burst photostimulation was chosen to mimic in vivo phasic firing of DA neurons (*Paladini and Roeper, 2014*). Single photostimulation-evoked EPSCs in both ChIs and SPNs (*Figure 3B*) were blocked by the AMPA-kainate receptor antagonist CNQX, confirming GLU mediation (n = 4 ChIs per genotype; n = 3 SPNs per genotype). As reported previously (*Chuhma et al., 2014*), the amplitude of EPSCs in ChIs (CTRL 51 ± 6.0 pA) was greater than in SPNs (CTRL 21.5 ± 2.2 pA). The amplitude of single-evoked EPSCs was unaffected in cHET mice (ChIs 66.1 ± 6.7 pA; SPNs 21.2 ± 2.3 pA) (*Figure 3B*), as were EPSC rise and decay time constants (*Figure 3—figure supplement 2F,G*). Burst-induced EPSCs in ChIs and SPNs showed short-term depression in both genotypes that was significantly greater in cHETs (*Figure 3C*). This was particularly evident when EPSC amplitudes were normalized to the first EPSC in the burst, which showed no genotypic difference (*Figure 3C*, graphs). In CTRL mice, EPSCs in ChIs decreased to 48 ± 6.0% with the second pulse and to 23 ± 4.2% with the fifth, while in cHET mice EPSCs decreased to 20 ± 6.3% with the second and to 14 ± 3.3% with the fifth. The rundown was apparently faster in SPNs (*Figure 3C*, bottom traces and graph); in CTRL mice, EPSCs decreased to 48 ± 6.0% with the second pulse and to 24 ± 6.4% with the fifth, while in cHET mice EPSC amplitude decreased to 25 ± 4.9% with the second, and to 16 ± 1% with the fifth, which was close to baseline. Observing a more rapid frequency-dependent EPSC depression in cHETs in both ChIs and SPNs, and no differences in their intrinsic properties (*Figure 3—figure supplement 2B,C,D,E*), is consistent with a presynaptic reduction in PAG. The average amplitude and frequency of spontaneous EPSCs, measured in both the SPNs and ChIs, showed no genotypic difference (*Figure 3—figure supplement 2H,I*), indicating that GLU inputs mostly from forebrain regions, as well as signaling through postsynaptic GLU receptors, was unaffected in DAT GLS1 cHETs.

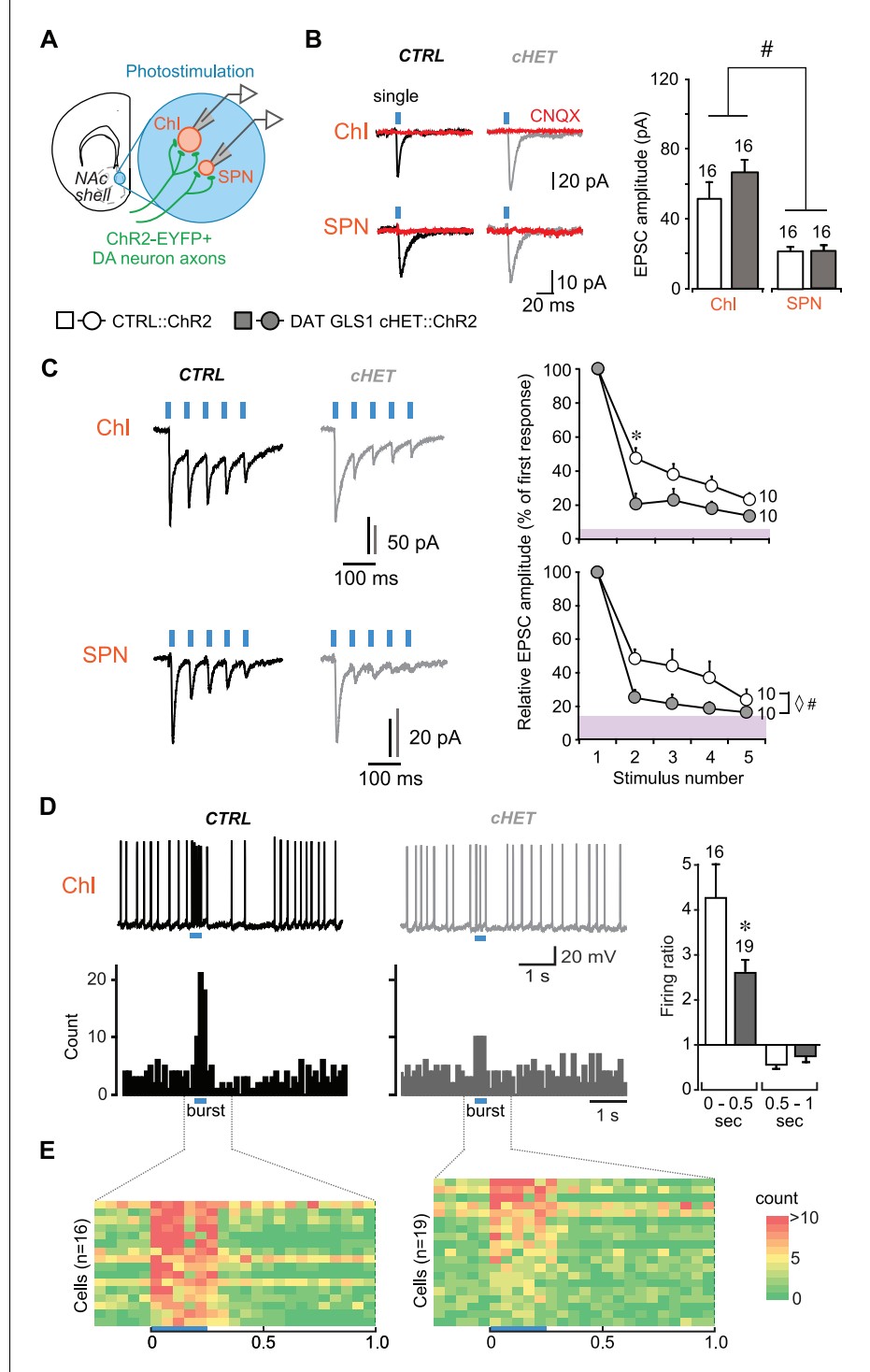

**Figure 3.** DA neuron GLU cotransmission is attenuated in DAT GLS1 cHETs at phasic firing frequencies. (**A**) Schematic of a coronal slice (−1.34 mm from bregma) indicating the location of the patch-clamp recordings in the medial NAc shell. DA neuron excitatory responses evoked by photostimulation (blue circles) were measured from ChIs and SPNs (left). See also *Figure 3—figure supplement 1*. (**B**) Representative traces (left) of EPSCs generated by a single-pulse photostimulation (blue bar) at 0.1 Hz recorded from ChIs and SPNs. Traces shown are averages of 10 consecutive traces. Comparison is made between responses in CTRL (black traces) and DAT GLS1 cHET mice (gray traces); all responses were completely blocked by CNQX (40 μM; red traces). Summary of average EPSC amplitude after single-pulse photostimulation (right). # indicates a significant main effect of cell type (two-
*Figure 3 continued on next page*

*Figure 3 continued*

way ANOVA, $F_{(1,36)}$ = 25.6, p<0.001, ES partial $\eta^2$ = 0.42); there was no significant genotype effect ($F_{(1,36)}$ = 1.084, p=0.305), nor interaction ($F_{(1,36)}$ = 0.628, p=0.433). See also *Figure 3—figure supplement 2*. (C) Representative traces of EPSCs generated by burst photostimulation (5 pulses at 20 Hz) recorded from ChIs (top) and SPNs (bottom). Summary of the average EPSC amplitudes after burst photostimulation (right) are shown as percentage of the first response, which did not differ between genotypes (ChIs, CTRL 95 ± 29 pA vs. cHET 107 ± 12 pA, Mann-Whitney, p=0.14; SPNs, CTRL 27 ± 4 pA vs. cHET 28 ± 5 pA, Mann-Whitney, p=0.88). The shaded violet bar at the bottom of the graphs represents the average baseline noise (ChIs 3.8 ± 0.4 pA; SPNs 3.5 ± 0.3 pA). For ChIs, repeated measures (RM) ANOVA revealed a significant pulses X genotype interaction ($F_{(3,54)}$ = 28.2, p=0.006, ES partial $\eta^2$ = 0.27), main effect of pulses ($F_{(3,54)}$ = 20.9, p<0.001), and main effect of genotype ($F_{(1,18)}$ = 5.06, p=0.037). * indicates significant difference from CTRL (p=0.006) after applying a Bonferroni correction for 4 comparisons ($\alpha$ = 0.0125). For SPNs, ◊ # indicates a significant main effect of genotype ($F_{(1,18)}$ = 4.6, p=0.047, ES partial $\eta^2$ = 0.20) and main effect of pulses ($F_{(3,54)}$ = 7.7, p<0.001, ES partial $\eta^2$ = 0.30) by RM ANOVA; but no significant interaction ($F_{(3,54)}$ = 2.0, p=0.101). (D) Effect of photostimulation mimicking DA neuron bursting (5 pulses at 20 Hz) on ChI firing. Representative traces are shown above (left), with peristimulus histograms summing ten consecutive traces (0.1 s bin) below. Ratio of firing during burst photostimulation (0–0.5 s from onset of train) and after (0.5–1 s from onset) to baseline firing are shown on the right. * indicates significant effect of genotype (one-way ANOVA, $F_{(1,33)}$ = 7.0, p=0.013, ES partial $\eta^2$ = 0.17). (E) Colored-coded tables showing action potential counts in 50 ms intervals, prior to, during and after DA terminal photostimulation for CTRL (left) and DAT GLS1 cHET mice (right) for all recorded cells. The blue horizontal bar at the bottom of each table indicates the duration of burst photostimulation, with onset at time 0. In all the graphs, the number of cells is shown above the bars or next to the lines. In this and subsequent figures, error bars represent SEM. See *Figure 3—source data 1*.xlsx for source data and statistical analysis.

The following source data and figure supplements are available for figure 3:

**Source data 1.** Slice patch clamp experiments.

**Figure supplement 1.** Comparison between CTRL::ChR2 and DAT GLS1 cHET::ChR2 mice showing selective ChR2 expression in DA neurons did not differ between genotypes.

**Figure supplement 2.** Comparison between CTRL::ChR2 and DAT GLS1 cHET::ChR2 mice showing that intrinsic electrophysiological membrane properties and spontaneous EPSCs measured in NAc shell cells did not differ between genotypes.

**Figure supplement 2—source data 1.** Intrinsic electrophysiological membrane properties and spontaneous EPSCs measured in NAc shell cells of CTRL and DAT GLS1 cHET mice .

At the striatal circuit level, DA neuron control of ChI firing in the medial NAc shell (*Chuhma et al., 2014*) was attenuated in cHET mice (*Figure 3D*). We quantified this using the firing ratio, the firing frequency during train photostimulation (0–0.5 s from the onset of train) divided by the preceding 2 s of baseline firing. There were no genotypic differences in baseline firing frequencies (CTRL 4.7 ± 1.1 Hz; cHET 3.9 ± 0.6 Hz; ANOVA, $F_{(1,33)}$= 0.60, p=0.444). The firing ratio in CTRL mice was 4.3 ± 0.7 compared to 2.1 ± 0.2 in cHET mice, which was significantly reduced (*Figure 3D*, right). In the subsequent half-second window, the firing ratio reversed to below baseline in CTRL (0.6 ± 0.08) and cHETs (0.7 ± 0.11), which did not differ (*Figure 3D*, right). This reduction in firing is mainly mediated by activity-dependent components, and less so by DA D2-mediated inhibition (*Chuhma et al., 2014*). Color-coded tables with a 50 msec window (*Figure 3E*) clearly show greater burst firing in CTRL than in cHET, but little difference in the post-burst period. Dividing the 0.5 to 1 s interval into 250 ms windows revealed no significant differences (one-way ANOVA: 0.5–0.75 period, CTRL 0.6 ± 0.12 vs. cHET 0.8 ± 0.12, $F_{(1,34)}$= 1.98, p=0.168; 0.75–1 period, CTRL 0.5 ± 0.08 vs. cHET 0.8 ± 0.1, $F_{(1,34)}$= 3.510, p=0.070). Thus, PAG plays an important role in sustaining DA neuron GLU cotransmission at higher firing frequencies and determines their ability to drive ChIs to fire in bursts.

## Normal DA transmission in DAT GLS1 cHETs

To evaluate the specificity of the reduction in GLU cotransmission in DAT GLS1 cHET mice further, we counted DA neurons by unbiased stereology, at P110 (*Figure 4A*). We found no reduction in the number of DA neurons in the VTA (unilateral counts: CTRL 7548 ± 418, cHET 7310 ± 450) or SNc (CTRL 6595 ± 373, cHET 6781 ± 518). DA neurons in cHET mice showed no differences in their intrinsic electrophysiological properties (*Figure 4—figure supplement 1*). Presynaptic DA content and turnover, in the NAc and dStr of adult mice (P71-P110), did not significantly differ between genotypes (*Figure 4B*). We performed fast-scan cyclic voltammetry (FSCV) in DAT GLS1 cHET::ChR2 mice (P71-P85) to determine whether DA release dynamics were affected (*Figure 4C*). We compared DA release evoked by single or burst photostimulation in the NAc medial shell. To challenge DA neuron synapses further, single pulse stimulation was repeated twice followed by a burst, and burst stimulation was repeated twice followed by a single. There were no genotypic differences in DA release with either stimulation pattern (*Figure 4D*). The decay time constant of DA responses did not differ significantly between genotypes with single (CTRL 409 ± 30 ms; cHET 362 ± 26 ms; ANOVA, $F_{(1,23)}$=1.42, p=0.245) or burst photostimulation (CTRL 540 ± 28 ms; cHET 482 ± 25 ms; ANOVA, $F_{(1,22)}$= 2.12, p=0.160). Thus, the conditional *Gls1* reduction does not affect DA neuron DA release in the NAc medial shell, where GLU cotransmission is strongest. Evoked DA release was not affected in the NAc core (*Figure 4—figure supplement 2A,B,C*) nor in the dStr (*Figure 4—figure supplement 2D,E,F*), indicating that DA storage and release dynamics throughout the striatum are normal in DAT GLS1 cHETs. The effect sizes for all non-significant F values were small to negligible (partial $\eta^2$: Stereology = 0.014; DA content = 0.004; DA release: range 0.0002 to 0.011). Thus, DA neuron development and DA transmission are unaffected in DAT GLS1 cHETs.

## DA neuron dependent behaviors unaffected in DAT GLS1 cHETs

We examined DA neuron dependent behaviors in DAT GLS1 cHETs (P90-120). We assessed motor learning and coordination on the rotarod, which is affected following neurotoxic loss of DA neurons (*Rozas et al., 1998*; *Beeler et al., 2010*) and also variably affected in DAT VGLUT2 cKO mice (*Fortin et al., 2012*) (but see also *Birgner et al., 2010*; *Hnasko et al., 2010*). DAT GLS1 cHET mice showed robust motor learning, which did not differ from CTRL mice, on the first training day, when rotarod speed was 20 rpm (*Figure 5A*), and then accelerated to 30 rpm and 40 rpm on subsequent days. Novelty-induced exploration in the open field was unaffected (*Figure 5B*). Mice used in this experiment belonged to two cohorts that were subsequently used in the amphetamine-induced locomotion and sensitization experiments. The results from the first cohort were replicated in the second cohort; since there was no significant cohort effect (two-way ANOVA for total locomotion in 60 min: cohort, $F_{(1,100)}$ = 50.9, p=0.64; cohort X genotype, $F_{(1,100)}$ = 2.6, p=0.11), the cohorts were combined.

DA neuron loss can have anxiogenic effects (*Drui et al., 2014*), and DAT VGLUT2 cKO mice showed decreased time spent in the center of the open field, indicative of increased anxiety (*Birgner et al., 2010*). DAT GLS1 cHET and CTRL mice spent the same time in the center of the open field (CTRL = 256± 43 s; DAT GLS1 cHET = 301 ± 23 s; one-way ANOVA, no genotype effect, $F_{(1,102)}$ = 0.551, p=0.46). We tested the mice in the elevated plus maze, another test of anxiety. A large cohort of mice (CTRL = 30 mice, cHET mice = 37 mice) was tested in an elevated plus maze with short arms. DAT GLS1 cHET and CTRL mice spent the same time in the open arms (CTRL = 31.7 ± 3.8 s, cHET = 25.4 ± 3.5 s; one-way ANOVA, no genotype effect, $F_{(1,65)}$=1.11, p=0.30). A small effect size of 0.022 (partial $\eta^2$) was detected. So, we tested a second cohort in a more anxiogenic elevated plus maze with longer arms (*Figure 5C*). We found no difference between genotypes in the time spent in the open arms, nor was there a difference between time spent in the open arms per entry (*Figure 5C*), or the time spent in the proximal and distal portions of the longer arms (proximal time, CTRL = 47.4 ± 4.4 s, cHET = 40.9 ± 4.9 s, one-way ANOVA, no genotype effect, $F_{(1,24)}$ = 0.887, p=0.36; distal time, CTRL = 43.7 ± 8.5 s, cHET = 51.27 ± 7.90 s, $F_{(1,24)}$ = 0.41, p=0.53).

DA neurons play a role in fear conditioning (*Fernandez Espejo, 2003*; *Wen et al., 2015*). Moreover, *stopGls1* HET mice, with a global *Gls1* reduction, show reduced contextual fear conditioning (*Gaisler-Salomon et al., 2009b*). However, DAT GLS1 cHET mice showed normal tone- and context-dependent fear conditioning (*Figure 5D*).

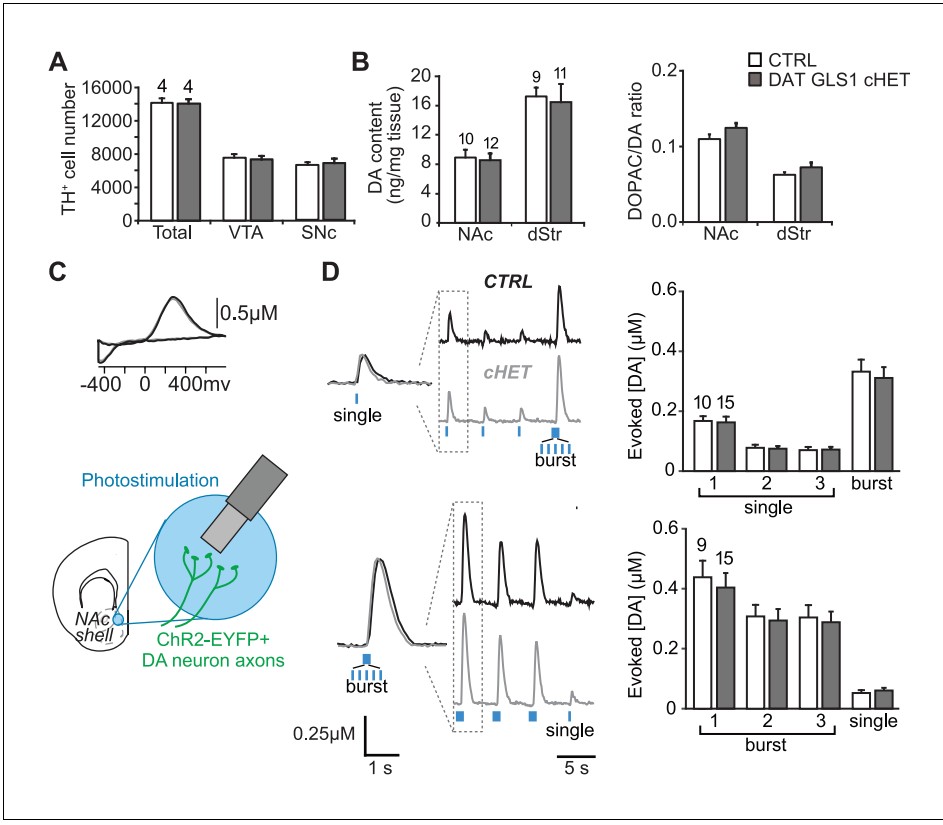

**Figure 4.** PAG reduction in DA neurons does not alter the number of DA neurons or striatal DA function. (**A**) Stereological-estimate of the total number of DA neurons (TH[+] neurons) in the VTA and SNc in one hemisphere showed no difference between genotypes (one-way ANOVA: VTA, $F_{(1,6)}$ = 0.149, p=0.713; SNc, $F_{(1,6)}$ = 0.085, p=0.781). There were no differences in DA neuron intrinsic electrophysiological properties; see ***Figure 4—figure supplement 1***. (**B**) Tissue DA content in the NAc and dStr (left) and DA turnover measured by DOPAC/DA ratio (right) did not differ between genotypes by one-way ANOVA (NAc DA content, $F_{(1,22)}$ = 0.070, p=0.794; NAc DOPAC/DA, $F_{(1,22)}$ = 3.01, p=0.098; dStr DA content, $F_{(1,20)}$ = 0.078, p=0.783; dStr DOPAC/DA, $F_{(1,20)}$ = 1.68, p=0.211). (**C**) FSCV recordings in the medial NAc shell. A representative voltammogram is shown above a schematic of a coronal slice (−1.34 mm from bregma) indicating the recording configuration. (**D**) DA release evoked by three consecutive single photostimulation pulses followed by a burst (5 pulses at 20 Hz) (above), or by three consecutive bursts followed by a single (below). Representative recordings of evoked DA release are shown with dashed boxes indicating initial traces that were enlarged and superimposed on the left, showing that DA release dynamics did not differ between genotypes for the single (above) or burst (below) responses. DA release dynamics did not differ between genotypes for consecutive singles followed by a burst (above) or repeated bursts followed by a single pulse (below). The average evoked DA release is shown on the graph (right). For consecutive single pulses followed by a burst, a RM ANOVA revealed a significant main effect of pulses ($F_{(3,69)}$ = 135.1, p<0.001, ES partial $\eta^2$ = 0.85); there was no effect of genotype ($F_{(1,23)}$ = 0.069, p=0.795) nor interaction ($F_{(3,69)}$ = 0.247, p=0.864). For the consecutive bursts followed by a single, a RM ANOVA revealed a significant main effect of pulses ($F_{(3,66)}$ = 124.5; p<0.001, ES partial $\eta^2$ = 0.85); there was no effect of genotype ($F_{(1,22)}$ = 0.004, p=0.948) or interaction ($F_{(3,66)}$ = 0.103, p=0.103). Dopamine release in the NAc core and dStr was also not affected in DAT GLS1 cHETs; see ***Figure 4—figure supplement 2***. Numbers of mice or the number of slices (FSCV) are shown in each graph above the bars. See ***Figure 4—source data 1***.xlsx for source data and statistical analysis.

The following source data and figure supplements are available for figure 4:

**Source data 1.** Dopamine transmission in CTRL and DAT GLS1 cHET mice.

**Figure supplement 1.** Electrophysiological properties of putative DA neurons in the ventral midbrain.

**Figure supplement 1—source data 1.** Dopamine neuron membrane properties in CTRL and DAT GLS1 cHET mice.

*Figure 4 continued on next page*

*Figure 4 continued*

**Figure supplement 2.** Dopamine release in nucleus accumbens core and dorsal striatum, measured by fast-scan cyclic voltammetry (FSCV), is not affected in DAT GLS1 cHET mice.

**Figure supplement 2—source data 2.** Dopamine release measured by FSCV in the NAc Core and dStr in CTRL and DAT GLS1 cHET mice.

DA neurons are the substrate for psychostimulant-induced behaviors (*Lüscher and Malenka, 2011*), and DAT VGLUT2 cKO mice show a blunted locomotor response to amphetamine (*Birgner et al., 2010*) and cocaine (*Hnasko et al., 2010*). stopGLS1 HET mice also show a reduced response to acute amphetamine (*Gaisler-Salomon et al., 2009b*), revealing a role of PAG in amphetamine-induced responses. DAT GLS1 cHET mice responded to low (2.5 mg/Kg) and high (5 mg/Kg) doses of amphetamine indistinguishably from CTRL mice (*Figure 5E*).

For all these behavioral experiments, effect sizes were negligible for nonsignificant F values (partial $\eta^2$: Rotarod genotype effect = 0.0002 and interaction = 0.0085; Open Field genotype effect = 0.0064 and interaction = 0.0032; Center Time genotype effect = 0.0053, Elevated Plus Maze genotype effect = 0.0002, Context Fear Conditioning genotype effect = 0.013, Acute Amphetamine genotype effect = 0.0010 and interaction = 0.0054). Tone fear conditioning did show a medium effect size (partial $\eta^2$ = 0.067), but a significant genotypic effect was not seen in a replication experiment (*Figure 6F*). Thus attenuation of phasic GLU cotransmission does not affect motor performance, exploratory behaviors, anxiety regulation, fear conditioning or responses to acute amphetamine, revealing that several DA neuron VGLUT2-dependent and PAG-dependent behaviors were normal in DAT GLS1 cHET mice.

## Reduced amphetamine sensitization and potentiated latent inhibition in DAT GLS1 cHET mice

stopGLS1 HET mice manifest a schizophrenia resilience phenotype characterized behaviorally by reduced amphetamine sensitization and potentiated LI (*Figure 6—figure supplement 1*, and *Gaisler-Salomon et al., 2009b*); as do *Δ*GLS1 HET mice, with a global *Gls1* reduction, generated by breeding floxGLS1 mice with mice expressing cre under the control of the ubiquitous tamoxifen-inducible ROSA26 promoter (*Figure 6—figure supplements 2* and *3*). The activity of DA neurons projecting to the NAc shell, the majority of which are capable of GLU cotransmission, play a crucial role in both amphetamine sensitization and LI (*Ikemoto, 2007*; *Nelson et al., 2011*), so we asked whether DAT GLS1 cHETs display similar behavioral phenotypes.

We tested DAT GLS1 cHET mice (P90-P120) for amphetamine sensitization, following the protocol schematized in *Figure 6A*. Two cohorts were tested, since there was no difference between the cohorts (ANOVA cohort effect: CTRL veh, $F_{(1,17)}$ = 0.37, p=0.872; cHET Veh, $F_{(1,15)}$ = 0.49, p=0.494; CTRL Amph, $F_{(1,19)}$ = 0.94, p=0.346; cHET Amph, $F_{(1,19)}$ = 3.752, p=0.068) they were combined. With daily amphetamine injections (2.5 mg/kg) over 5 days, CTRL mice showed an increase in drug-induced hyperlocomotion, characteristic of a sensitized response (*Figure 6B*), while cHET mice showed no increase in hyperlocomotion. Ten days later, all mice were tested, first with a vehicle challenge (Day 18) and then with amphetamine (2.5 mg/kg; Day 19). The vehicle challenge revealed a modest but significant conditioned response in the Amph-treated groups. During the amphetamine challenge, amphetamine-treated CTRL mice showed a significant sensitized response, while amphetamine-treated cHET mice showed a significant but smaller sensitized response (*Figure 6B*, gray area). Further comparison of the locomotor response during the 90 min post-amphetamine (*Figure 6C*) showed no difference between vehicle-treated cHET and CTRL mice, but a significantly smaller sensitized response in amphetamine-treated cHET mice compared to amphetamine-treated CTRL mice. Thus attenuating phasic GLU cotransmission blocks the induction of amphetamine sensitization and reduces the expression of sensitization, after a withdrawal period.

LI is characterized by an attenuated response to a conditioned stimulus (CS) presented without reinforcement prior to being paired with an unconditioned stimulus (US) (*Weiner, 2003*). LI is

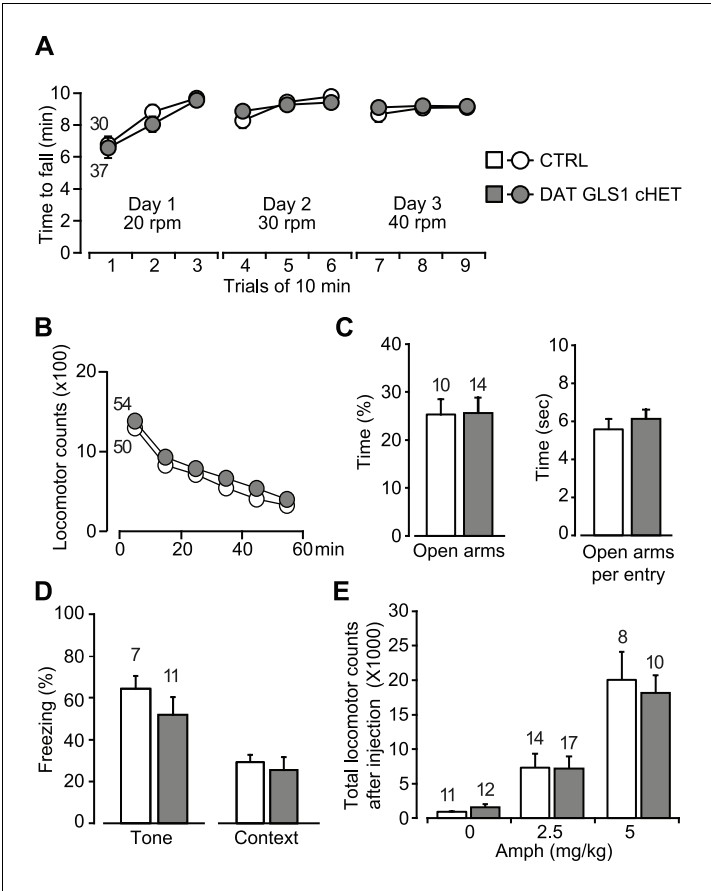

**Figure 5.** Motor performance, anxiety and amphetamine-induced hyperlocomotion are unaffected in DAT GLS1 cHETs. (**A**) Motor performance on an accelerating rotarod over 3 days showed no difference between genotypes (RM ANOVA, significant effect of trials, $F_{(8,520)} = 22.9$, p<0.0001, ES partial $\eta^2 = 0.26$); there was no effect of genotype ($F_{(1,65)} = 0.018$, p=0.894; nor interaction $F_{(8,520)} = 0.562$, p=0.809). (**B**) Locomotor activity in the open field for one hour revealed no genotypic difference in novelty-induced locomotion and habituation (RM ANOVA, main effect of time, $F_{(5,510)} = 193.0$, p<0.0001, ES partial $\eta^2 = 0.65$); no effect of genotype ($F_{(1,102)} = 0.664$, p=0.417) nor interaction ($F_{(5,510)} = 0.329$, p=0.895). (**C**) Exploration in the elevated-plus maze (5 min) showed no genotypic difference in percentage of time spent in the open arms (left) (one way-ANOVA, $F_{(1,22)} = 0.004$, p=0.949) or time spent in the open arms per entry (right) (one way-ANOVA, $F_{(1,22)} = 0.547$, p=0.467). (**D**) Fear conditioning to tone (left) measured as the average percentage of freezing during the CS (two tone presentations) or to a context previously paired with a shock (right) showed no genotypic differences (one-way ANOVA, tone fear conditioning, $F_{(1,16)} = 1.145$, p=0.300; context fear conditioning, $F_{(1,16)} = 0.207$, p=0.655). (**E**) Amphetamine-induced locomotor activity recorded over 90 min post injection showed no genotypic difference in the dose-dependent responses (two-way ANOVA, main effect of drug treatment, $F_{(2.66)} = 34.8$, p<0.0001, ES partial $\eta^2 = 0.51$; no effect of genotype, $F_{(2.66)} = 0.068$, p=0.795; nor interaction, $F_{(2.66)} = 0.18$, p=0.836). The number of mice is shown in the graphs above the bars or next to the lines. See *Figure 5—source data 1*.xlsx for source data and statistical analysis.

The following source data is available for figure 5:

**Source data 1.** Dopamine neuron dependent behaviors in CTRL and DAT GLS1 cHET mice.

potentiated by neurotoxin-induced loss of DA neurons projecting to the NAc shell (*Joseph et al., 2000*; *Nelson et al., 2011*), which would affect GLU cotransmission. We asked whether DAT GLS1 cHETs show potentiated LI, using the protocol schematized in *Figure 6D*. On Day 1, mice (P90-120) were assigned either to a preexposure (PE) group that received 20 tone exposures prior to tone (CS) - shock (US) pairing, or to a non-preexposure (NPE) group that received only the CS-US pairing. The number of CS pre-exposures was limited so as not to elicit LI in the PE group, enabling detection of

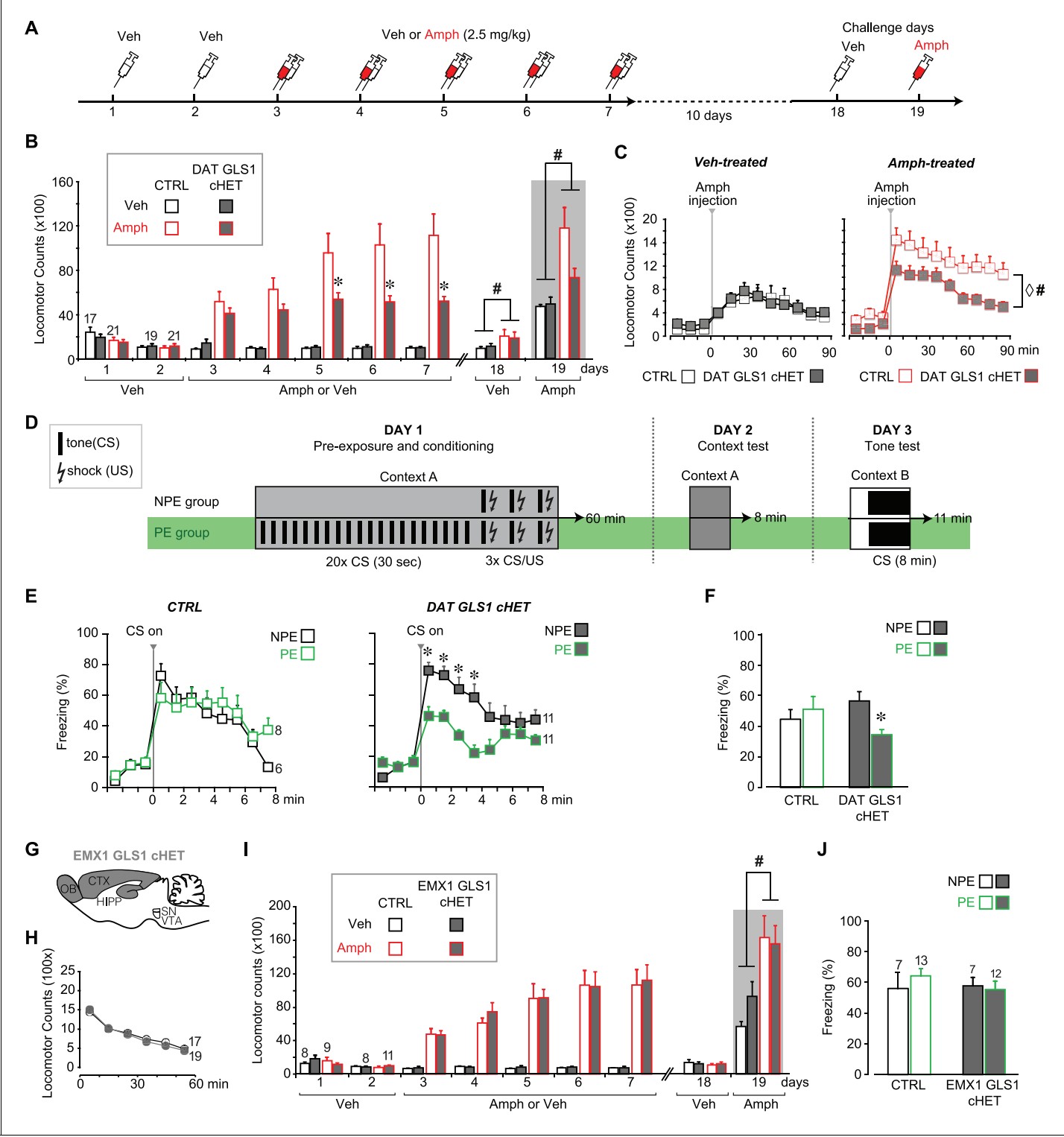

**Figure 6.** DAT GLS1 cHET mice showed attenuated amphetamine sensitization and potentiated latent inhibition. (**A**) Schematic of amphetamine sensitization protocol. (**B**) Locomotor activity in the open field after vehicle (Veh) or Amphetamine (Amph) injection. There were no between group differences in activity on the habituation days (Days 1 and 2). Over the subsequent 5 treatment days, CTRL mice showed sensitization to Amph while DAT GLS1 cHET mice did not (RM ANOVA, significant genotype X treatment X day interaction, $F_{(4,296)}$ = 4.4, p=0.002, ES partial $\eta^2$ = 0.06; RM ANOVA within Amph-treated mice, significant genotype X day interaction, $F_{(4,160)}$ = 5.9, p<0.001, ES partial $\eta^2$ = 0.112). *p<0.016 indicates significantly different from CTRL Amph-treated mice, after Bonferroni correction for 3 comparisons ($\alpha$ = 0.016). On the Veh challenge day (day 18), Amph-treated mice showed a modest increase in locomotion relative to Veh-treated mice independent of genotype. # indicates significant treatment effect ($F_{(1,74)}$= 4.03,

*Figure 6 continued on next page*

*Figure 6 continued*

p=0.048; partial $\eta^2$ = 0.052), but no main effect of genotype ($F_{(1,74)}$< 0.001, p=1) or significant interaction ($F_{(1,74)}$= 0.163, p=0.688). On the challenge day (Day 19), Amph-treated mice showed increased locomotion relative to Veh-treated mice independent of genotype. # indicates significant treatment effect (two-way ANOVA: $F_{(1,74)}$ = 13.7, p<0.001, ES partial $\eta^2$ = 0.112), with no significant genotype effect ($F_{(1,74)}$ = 2.76, p=0.101), but a trend for interaction ($F_{(1,74)}$ = 3.18, p=0.078). (C) On the Amph challenge day Veh-treated (left) and Amph-treated mice (right) received Amph and activity was monitored for 90 min. Veh-treated mice showed no genotypic difference in their response to Amph (RM ANOVA genotype effect, $F_{(1,74)}$ = 0.012, p=0.91; genotype X time interaction, $F_{(1,74)}$ = 0.53, p=0.83). Amph-treated CTRL mice showed a sensitized response to Amph while DAT GLS1 cHET did not. ◇ # indicate a significant genotype difference (RM ANOVA, $F_{(1,40)}$ = 89.3, p=0.034, ES partial $\eta^2$ = 0.107), and significant effect of time ($F_{(8,320)}$ = 12.8, p<0.0001, ES partial $\eta^2$ = 0.243), but no significant interaction ($F_{(8,320)}$ = 0.576, p=0.798). stopGLS1 mice, with a global GLS1 HET reduction, show attenuated amphetamine sensitization; see *Figure 6—figure supplement 1*. ΔGLS1 HET mice, generated by breeding floxGLS1 mice with mice expressing cre under the control of the ubiquitous tamoxifen-inducible ROSA26 promoter (*Figure 6—figure supplement 2*), also show attenuated amphetamine sensitization (*Figure 6—figure supplement 3*). (D) Schematic of latent inhibition protocol. (E) On the tone test day (Day 3), the percent time freezing for the 3 min before and 8 min after CS (tone) presentation are shown for CTRL (left) and DAT GLS1 cHET mice (right). CTRL non-preexposure (NPE) and preexposure (PE) groups did not differ, evidencing no LI (RM ANOVA during CS, no preexposure effect, $F_{(2,12)}$ = 0.127, p=0.728; nor preexposure X time interaction, $F_{(7,84)}$ = 1.66, p=0.129). DAT GLS1 cHET NPE and PE groups did not differ before CS presentation (PE effect, $F_{(1,20)}$ = 0.646, p=0.431; interaction, $F_{(2,40)}$ = 2.12, p=0.132); during CS presentation, PE mice showed less freezing than NPE mice, evidencing potentiated LI (RM ANOVA, significant time X PE treatment interaction, $F_{(7,140)}$= 2.88, p=0.008, ES partial $\eta^2$ = 0.126). *p<0.006 indicates significant different between PE and NPE groups, after Bonferroni correction for 8 comparisons (α = 0.006). (F) Percent total time freezing during 8 min CS presentation on the tone test (Day 3). DAT GLS1 cHET PE mice, but not CTRL mice, showed less freezing during CS presentation, evidencing potentiated LI (two-way ANOVA, significant genotype X PE treatment interaction, $F_{(1,32)}$ = 5.3, p=0.028, ES partial $\eta^2$ = 0.334; no significant genotype effect, $F_{(1,32)}$ = 0.145, p=0.71, nor PE effect, $F_{(1,32)}$ = 1.52, p=0.227). Within the NPE group, there was no genotype effect, showing that learning was not affected in DAT GLS1 cHETs ($F_{(1,15)}$= 1.56, p=0.23). * indicates significant pre-exposure effect within the DAT GLS1 cHET group by ANOVA ($F_{(1,20)}$ = 10.03, p=0.005, ES partial $\eta^2$ = 0.334). stopGLS1 mice (*Gaisler-Salomon et al., 2009b*), as well as ΔGLS1 HET mice (*Figure 6—figure supplement 3*), both with a global GLS1 reduction, show potentiation of LI. (G) Schematic of the EMX1 GLS1 cHET mouse brain (sagittal view) illustrating the GLS1 cHET genotype in forebrain. See *Figure 6—figure supplement 4*. (H) Novelty-induced locomotion and habituation to the open field did not differ between CTRL (white circles) and EMX1 GLS1 cHET mice (grey circles). RM ANOVA showed a significant time effect ($F_{(5,170)}$ = 138.1, p<0.0001, ES partial $\eta^2$ = 0.802); no significant genotype effect ($F_{(1,34)}$ = 0.599, p=0.44); and no significant interaction ($F_{(5,170)}$ = 0.820, p=0.537). (I) Both CTRL and EMX1 GLS1 cHET mice showed sensitization to Amph during the 5 treatment days (RM ANOVA: days X drug treatment effect, $F_{(4,128)}$= 11.33, p<0.0001, ES partial $\eta^2$ = 0.259; there was no significant day X drug treatment X genotype interaction, $F_{(4,128)}$= 0.161, p=0.96). On the Veh challenge day, there were no significant differences between genotypes of drug-treatment groups. On the Amph challenge day, Amph-treated mice showed a sensitized response relative to Veh-injected mice, independent of genotype. # indicates a significant main effect of drug treatment ($F_{(1,32)}$ = 16.83, p<0.0001, ES partial $\eta^2$ = 0.330). (J) EMX1 GLS1 cHET mice did not show potentiation of LI. Percent time freezing during the 8 min CS presentation on the tone test day (Day 3) did not differ between NPE and PE groups, independent of genotype (two-way ANOVA: no significant main effect of genotype, $F_{(1,35)}$ = 0.281, p=0.60; PE, $F_{(1,35)}$ = 0.163, p=0.69; or interaction, $F_{(1,35)}$= 0.586, p=0.45). EMX1 GLS1 cHET mice, as well as ΔGLS1 HET and DAT GLS1 cHET mice, showed clozapine-induced potentiation of LI (*Figure 6—figure supplement 5*). In all graphs, the number of mice is shown above the bars or next to the lines. See *Figure 6—source data 1*.xlsx for source data and statistical analysis.

The following source data and figure supplements are available for figure 6:

**Source data 1.** Amphetamine sensitization and latent inhibition in DAT GLS1 cHET and EMX1 GLS1 cHET mice.

**Figure supplement 1.** stopGLS1 HET with a global PAG reduction show attenuated amphetamine sensitization.

**Figure supplement 1—source data 1.** Amphetamine sensitization in stopGLS1 HET mice .

**Figure supplement 2.** Breeding ΔGLS1 HET mice (with a global GLS1 reduction) from floxGLS1 mice.

**Figure supplement 2—source data 2.** PAG protein determinations in Rosa26$^{ERT2cre}$ GLS1 mice.

**Figure supplement 3.** ΔGLS1 HET mice show reduced novelty-induced locomotion, attenuated amphetamine sensitization and potentiated latent inhibition.

**Figure supplement 3—source data 3.** Novelty-induced locomotion and amphetamine sensitization in ΔGLS1 HET mice.

**Figure supplement 4.** Conditional forebrain PAG reduction in EMX1 GLS1 cHET mice.

**Figure supplement 4—source data 4.** PAG protein determinations in EMX1 GLS1 cHET mice.

**Figure supplement 5.** Clozapine-induced potentiation of latent inhibition in EMX1 GLS1 cHET, ΔGLS1 HET and DAT GLS1 cHET mice.

*Figure 6 continued on next page*

*Figure 6 continued*

**Figure supplement 5—source data 5.** Clozapine-induced potentiation of latent inhibition.

potentiated LI. On Day 2, freezing to context was tested in the same chamber; there was no genotypic difference between the NPE and PE groups (CTRL NPE = 20 ± 4.5 s; cHET NPE = 35 ± 5.4 s; CTRL PE = 39.9 ± 5.9 s; cHET PE = 36.8 ± 5.7 s; two-way ANOVA; genotype factor, $F_{(1,32)}$ = 1.00, p=0.323; preexposure factor, $F_{(1,32)}$ = 3.46, p=0.074; interaction, $F_{(1,32)}$ = 2.358, p=0.134). On Day 3, mice were put in a different context and presented with the CS. Less freezing during CS presentation in PE compared to NPE groups reflects potentiation of LI. During the 3 min before CS presentation, both CTRL and cHET mice showed less than 20% freezing, and there was no difference between the NPE and PE groups (*Figure 6E*). During CS presentation, CTRL mice showed increased freezing with no difference between the NPE and PE groups (*Figure 6E*, left graph), revealing the learned tone-fear association and no LI. In contrast, the cHET PE group showed less freezing in comparison to the NPE group, revealing potentiated LI (*Figure 6E*, right graph). Importantly, when analyzing the total freezing during the CS presentation and comparing responses between genotypes directly, the cHET NPE group did not differ from the CTRL NPE group, showing that aversive associative learning per se was not affected in cHETs (*Figure 6F*), replicating previous findings (*Figure 5D*). Thus, the restricted *Gls1* reduction in DA neurons is sufficient to reduce amphetamine sensitization and potentiate LI.

## Normal amphetamine sensitization and no potentiation of latent inhibition in EMX1 GLS1 cHET mice

It is striking that the behavioral phenotypes seen in GLS1 HETs were engendered by the restricted *Gls1* reduction in DA neurons, and apparently do not depend on *Gls1* reductions in forebrain where GLS1 mRNA and PAG are highly expressed (*Kaneko, 2000*; *Gaisler-Salomon et al., 2012*). To verify this, we made a forebrain-restricted *Gls1* reduction by breeding EMX1$^{IREScre}$ mice with floxGLS1 mice to generate EMX1 GLS1 cHET progeny (*Figure 6G* and *Figure 6—figure supplement 4*). EMX1 GLS1 cHETs (P85-107) did not differ from CTRL mice in their novelty-induced locomotion in the open field (*Figure 6H*) and amphetamine sensitization (*Figure 6I*). The effect size for the nonsignificant drug treatment X time X genotype interaction was negligible (partial $\eta^2$: 0.005). EMX1 GLS1 cHETs (P80-96) did not show potentiation of LI (ES for nonsignificant PE X genotype interaction = 0.016) (*Figure 6J*). To confirm in EMX1 GLS1 cHETs that the limited number of pre-exposures did not elicit LI and yet was sufficient to reveal potentiation of LI, we tested for clozapine-induced potentiation of LI (*Gaisler-Salomon et al., 2009b*) (*Figure 6—figure supplement 5A*). In both CTRL and EMX1 GLS1 cHETs, clozapine treatment on Day 1, potentiated LI in the PE groups (*Figure 6—figure supplement 5B*), but had no effect in the NPE groups showing that it did not affect learning. Similar clozapine effects were seen in ΔGLS1 HET and DAT GLS1 cHET mice (*Figure 6—figure supplement 5C*). The lack of further potentiation of LI in ΔGLS1 HET and DAT GLS1 cHET mice suggests that clozapine treatment and *Gls1* deficiency in DA neurons each either induce maximal potentiation of LI, or involve shared mechanisms so that *Gls1* deficiency occludes clozapine-induced potentiation of LI. In summary, our results argue that reducing *Gls1* in DA neurons is not only sufficient but also necessary for the reduction of amphetamine sensitization and potentiation of LI.

## Discussion

Here we show that a conditional heterozygous reduction of *Gls1* in DA neurons selectively attenuates GLU cotransmission at phasic firing frequencies without directly affecting DA transmission, enabling a focus on the role of DA neuron GLU cotransmission. The conditional *Gls1* reduction in DAT GLS1 cHETs is extremely restricted as it affects only those DA neurons that express *Gls1* and also VGLUT2 (about one third of VTA neurons and one tenth of SN neurons) and are thus capable of GLU cotransmission. The conditional *Gls1* reduction attenuates DA neuron excitatory drive in a frequency-dependent manner, further adding to its restricted impact, and revealing a crucial role of

PAG in GLU cotransmission. Strikingly, this modest *Gls1* heterozygous reduction profoundly affects two DA neuron dependent behaviors, namely psychostimulant sensitization and LI (*Table 1*), suggesting that phasic GLU cotransmission regulates attribution of motivational salience. The affected behaviors are components of the schizophrenia resilience profile of global GLS1 HETs and align with the actions of antipsychotic drugs, revealing that potential therapeutic effects of PAG inhibition may be mediated by attenuated DA neuron GLU cotransmission.

## PAG in DA neurons supports GLU cotransmission during sustained firing

A stereological analysis of PAG expression in DA neurons revealed that about half of DA neurons express PAG in both the VTA and SNc, in contrast to VGLUT2 expression, which is mostly restricted to DA neurons in the VTA (*Yamaguchi et al., 2015*). The function of PAG in SNc DA neurons incapable of GLU cotransmission is still uncertain, although we show that a minor reduction of PAG expression in those neurons had no impact on their survival or intrinsic physiology, nor did it affect DA transmission in the dStr or motor behaviors controlled by the dStr. In contrast, DA neurons capable of GLU cotransmission (TH[+] / VGLUT2[+] cells) preferentially express PAG, and a reduction of PAG in those VTA DA neurons was sufficient to attenuate phasic GLU cotransmission in the NAc shell and impact behaviors controlled by the NAc, revealing the important role of PAG in DA neuron GLU cotransmission.

The reduction in PAG activity of about 20% seen in stopGLS1 HET brain slices (*El Hage et al., 2012*) is associated with about a 15% reduction in GLU content (*Gaisler-Salomon et al., 2009b*) that presumably reflects a presynaptic diminution, since the highest concentrations of GLU are intracellular (*Danbolt, 2001*). Decreases in presynaptic GLU lead to decreases in vesicular GLU content and synaptic efficacy (*Ishikawa et al., 2002*). In DAT GLS1 cHETs, the first EPSC elicited by burst photostimulation was unaffected, as was observed in cultured GLS1 KO neurons (*Masson et al., 2006*), indicating that the readily releasable vesicle pool is replete. Smaller subsequent responses may reflect either diminished filling of the recycling pool (*Alabi and Tsien, 2012*), or decreased probability of release of vesicles with diminished GLU content (*Iwasaki and Takahashi, 2001*).

PAG expression in VTA DA neurons is weak to moderate relative to other brain regions (*Kaneko, 2000*). The fact that a heterozygous GLS1 reduction in DA neurons is sufficient to decrease synaptic efficacy indicates that PAG levels are not only lower but rate limiting. Single cell RT-PCR studies show that DA neurons also have low VGLUT2 mRNA copy numbers (*Trudeau et al., 2014*). Lower VGLUT2 expression would place further demands on the GLU-glutamine cycle to sustain synaptic transmission during periods of high activity, given that vesicular loading depends both on cytosolic GLU concentration and vesicular transporter number (*Wilson et al., 2005*). This indicates that the DA neuron GLU cotransmission in DAT GLS1 cHETS is highly dependent on PAG activity, and suggests that the global reduction in PAG activity in global GLS1 HETs affects DA neuron GLU cotransmission preferentially.

**Table 1.** Behaviors affected in DAT GLS1 cHET mice.

| Psychological domain | Behavioral test | Result |
|---|---|---|
| Motor skills and exploration | Rotarod | — |
| | Novelty-induced locomotion | — |
| Anxiety | Open field – center time | — |
| | Elevated plus maze | — |
| Associative learning | Fear conditioning | — |
| Psychostimulant response | Acute Amph-induced hyperlocomotion | — |
| | Amph sensitization | Reduced |
| Attention | Latent inhibition | Potentiated |

## Role of DA neuron glutamate cotransmission

Discerning the behavioral role of DA neuron GLU cotransmission has been challenging because of the impact of knocking out VGLUT2 in DA neurons on DA function. In DAT VGLUT2 cKOs, DA neuron function is affected profoundly due to the developmental role of VGLUT2 in DA neurons (*Fortin et al., 2012*). In DAT GLS1 cHETs, DA neuron DA functions appear normal; *Gls1* reduction affects neither the survival of DA neurons nor their intrinsic electrophysiological properties. VGLUT2 also plays an important role in vesicular DA uptake (*Hnasko et al., 2010*), but there was no impact of *Gls1* deficiency on DA content or release, even when DA terminals were stimulated repeatedly to increase the demand on DA release. Since DAT GLS1 cHET DA neurons show normal GLU cotransmission with low-frequency activity, our results suggest that modestly reduced presynaptic GLU is sufficient for the maintenance of normal vesicular DA dynamics in adulthood. Alternately, DA neuron GLU release may arise from segregated release sites (*Zhang et al., 2015*), so reduced vesicular GLU filling would not affect DA release. In the absence of a direct effect on synaptic DA transmission, finding that GLU signaling with high-frequency activity was affected selectively in DAT GLS1 cHETs allowed us to focus on the function of GLU cotransmission.

DAT GLS1 cHET mice do not show several behavioral phenotypes of DAT VGLUT2 cKOs, such as decreased novelty-induced locomotion, motor deficits on the rotarod, an anxiety phenotype, or blunted responses to psychostimulants (*Birgner et al., 2010*; *Hnasko et al., 2010*; *Fortin et al., 2012*). Presumably the behaviors not affected by a mild disruption in GLU cotransmission, are sensitive to manipulations that affect both DA and GLU transmission, such as in DAT VGLUT2 cKO mice. Strikingly, the subtle activity-dependent reduction in DA neuron GLU cotransmission in DAT GLS1 cHETs had major effects on amphetamine sensitization and LI, arguing that DA neuron GLU cotransmission is a key regulator of these behaviors. DA signaling increases with psychostimulant sensitization (*Vezina, 2004*; *Bocklisch et al., 2013*; *Covey et al., 2014*). While DA neuron DA signaling was not affected in DAT GLS1 cHETs, changes in DA signaling with repeated psychostimulant administration are likely, although attenuated due to reduced DA neuron GLU cotransmission. DA neuron excitatory connections to SPNs in the NAc core are modestly but significantly increased weeks after chronic psychostimulant administration (*Ishikawa et al., 2013*); psychostimulant-induced plasticity may be even stronger at DA neuron excitatory connections to ChIs in the NAc shell (*Chuhma et al., 2014*). At the VTA-NAc circuit level, reducing GLU cotransmission may attenuate increases in DA neuron activity associated with sensitization (*Bocklisch et al., 2013*). While subsequent circuit effects impacted by reduced GLU cotransmission involve DA signaling, we show here for the first time that the attenuation of GLU cotransmission in the absence of developmental alterations and direct effects on DA transmission has strong and selective behavioral effects, revealing a new mechanism through which DA neurons control behavior.

## Implications for salience and schizophrenia-resilience

DA neuron activity mediates both amphetamine sensitization and LI by encoding motivational salience of relevant events (*Young et al., 2005*; *Bromberg-Martin et al., 2010*; *Robinson et al., 2016*). Our results suggest that DA neuron GLU signaling plays a key role in salience attribution. In amphetamine sensitization, increases in DA neuron firing are restricted to medial VTA DA neurons (*Lodge and Grace, 2012*), the majority of which are capable of GLU cotransmission (*Yamaguchi et al., 2015*). Recent evidence suggests that all abused drugs increase DA neuron activity to strengthen the motivational salience of drug exposure or associated events (*Covey et al., 2014*). DAT GLS1 cHET mice do not show sensitization to amphetamine with repeated administration, and after a withdrawal period show reduced expression of sensitization. Similar results were found in mice with a conditional NR1 deletion in their DA neurons that resulted in a dramatic reduction in phasic firing (*Zweifel et al., 2009*). While the development of sensitization was unaffected in DAT NR1 cKO mice, the mice showed reduced expression of sensitization weeks after withdrawal (*Zweifel et al., 2008*). Taken together, several lines of evidence suggest that phasic DA neuron GLU signals facilitate sensitization by determining how rapidly and efficiently pathological levels of salience are attributed to drug exposure. In contrast, DAT NR1 cKO showed a reduction in conditioned responses to context not seen in the present study, suggesting that the abrogation of both DA and GLU phasic transmission must be affected to impact the development of drug-induced conditioned responses.

In LI, it is thought that the activity of DA neurons in the NAc updates the salience of a preexposed stimulus during the conditioning phase by integrating previous with current behavioral experiences (*Young et al., 2005*). Thus, the potentiation of LI seen in DAT GLS1 cHET mice represents a failure of DA neurons to increase the salience of the inconsequential preexposed stimulus under changed reinforcement contingencies during conditioning. The temporal precision of the DA neuron GLU signal makes it particularly suitable for updating salience. In the NAc medial shell, a structure known to regulate motivational salience (*Ikemoto, 2007*), DA neuron GLU connections to ChIs drive them to fire in bursts (*Chuhma et al., 2014*). Direct optogenetic excitation of ChIs in the NAc shell — as would result from DA neuron GLU actions — does not drive reinforcement learning on its own but instead modulates learning (*Lee et al., 2016*). It has been recently shown that GLU cotransmission is not required for self-administration reinforced by DA neuron activation (*Wang et al., 2017*). Taken together with our behavioral results, this suggests that DA neuron GLU signals modulate learning by regulating the attribution of motivational salience to relevant events via their direct control of ChI activity.

In the context of schizophrenia, the behaviors affected in DAT GLS1 cHETs align with the schizophrenia resilience phenotype of stopGLS1 HET (*Gaisler-Salomon et al., 2009b*), as well as ΔGLS1 HET mice, both with a global GLS1 heterozygous reduction. Several other phenotypes of GLS1 HETs, a blunted locomotor response to novelty, diminished sensitivity to acute amphetamine or reduced contextual fear conditioning were not seen in DAT GLS1 cHETs and so apparently do not depend on GLU cotransmission. Furthermore, none of these behavioral deficits were recapitulated in EMX1 GLS1 cHETs, with a forebrain-restricted GLS1 reduction, demonstrating that PAG in DA neurons is necessary for amphetamine sensitization and potentiation of LI, and reinforcing the likelihood that DA neuron GLU cotransmission is particularly sensitive to PAG reduction.

Modeling resilience in mice using transgenic approaches offers a direct path to intervention, as resilience mutations point directly to therapeutic targets (*Mihali et al., 2012*). Supported by the recent demonstration of VGLUT2 expression — and thus of GLU cotransmission — in primate DA neurons (*Root et al., 2016*), the therapeutic potential of PAG inhibition as a pharmacotherapy for schizophrenia (*Mingote et al., 2016*) may involve tempering DA neuron GLU cotransmission. Finally, our findings put forward the possibility that an increase in GLU cotransmission in the NAc may contribute to the pathophysiology of schizophrenia, in particular to aberrant salience leading to psychosis. Increased activity in the midbrain and NAc has been associated with aberrant salience attribution to irrelevant stimuli in patients with psychosis or individuals at high risk (*Romaniuk et al., 2010*; *Roiser et al., 2013*), while increased NAc activity does not correlate with increased dopamine synthesis capacity (*Roiser et al., 2013*). This inconsistency would be reconciled if increased activity in the VTA and NAc in SCZ is associated with a greater pathological increase in GLU cotransmission than in DA transmission.

## Materials and methods

### Experimental animals

We used stopGLS1 (JIMSR Cat# JAX:017956, RRID:IMSR_JAX:017956, *Gaisler-Salomon et al., 2009b*) and floxGLS1 mice (IMSR Cat# JAX:017894, RRID:IMSR_JAX:017894, *Mingote et al., 2016*), both on a 129SVE-F background, and DAT$^{IREScre}$ (RRID:IMSR_JAX:006660), EMX1$^{IREScre}$ (RRID:IMSR_JAX:005628) and Rosa26$^{creERT2}$ mice (RRID:IMSR_JAX:008463) on a C57BL/6 background. These mice were used to generate DAT GLS1 cHET or cKO mice, EMX1 GLS1 cHET or cKO mice, and ΔGLS1 HET mice, all on a mixed 129SVE-F and C57BL/6 background. For optogenetic stimulation, mice were bred with Ai32 (RCL-ChR2(H134R)/EYFP) mice (RRID:IMSR_JAX:024109) mice to confer conditional expression of ChR2 in their DA neurons. Inducible Rosa26$^{creERT2}$::GLS1$^{lox/+}$ mice were used to produce a global heterozygous GLS1 deletion in adulthood by administration of tamoxifen. Tamoxifen (Sigma-Aldrich, T5648) was dissolved in a peanut oil/ethanol (9:1 mixture) at 25 mg/ml, solubilized by vortexing for 5 min and warming to 37°C for several hours. Mice received 0.2 mL i.p. (5 mg Tamoxifen) daily for 5 successive days. Tamoxifen-treated Rosa26$^{creERT2}$::GLS1$^{Δ/+}$ mice were then crossed with wild-type C57BL/6 mice (RRID:IMSR_JAX:000664) to generate ΔGLS1 HETs.

## Immunohistochemistry

Mice were anesthetized with ketamine (90 mg/kg) + xylazine (7 mg/kg) and perfused with cold PBS followed by 4% paraformaldehyde (PFA), the brains removed, post-fixed overnight in 4% PFA, and cut at 50 μm with a vibrating microtome (Leica VT1200S). Coronal slices were collected into a cryo-protectant solution (30% glycerol, 30% ethylene glycol in 0.1 M Tris HCl [pH 7.4]) and kept at −20°C until processing. Sections were washed in PBS (100 mM; pH 7.4) and incubated in glycine (100 mM) for 30 min to quench aldehydes. Non-specific binding was blocked with 10% normal goat serum (NGS; Millipore) in 0.1% PBS Triton X-100 for 2 hr (PBS-T). Primary antibodies used were anti-TH (1:10,000 dilution, mouse monoclonal, Millipore Cat# MAB318 RRID:AB_2201528), anti-PAG (1:10,000 dilution, rabbit polyclonal, Norman Curthoys, Colorado State), and anti-GFP (1:2000 dilution; rabbit polyclonal, Millipore Cat# AB3080 - RRID:AB_91337). Secondary antibodies were: anti-rabbit Alexa Fluor 488 (1:200 dilution, ThermoFisher Scientific Cat# A-21206 RRID:AB_2535792) and anti-mouse Alexa Fluor 594 (ThermoFisher Scientific Cat# A-21203 RRID:AB_2535789). Primary antibodies in 0.02% PBS-T and 2% NGS were applied for 24 hr at 4°C. Sections were then washed with PBS and secondary antibodies applied for 45 min in 0.02% PBS-T at room temperature. Sections were mounted on slides and cover slipped with Prolong Gold aqueous medium (ThermoFisher Scientific) and stored at 4°C. Fluorescence images were acquired with a Fluoview FV1000 (Olympus) or A1 (Nikon) confocal laser scanning microscope, or a Axiovert 35M (Zeiss) epifluorescence microscope.

## Stereological analysis of DA neuron number

The SNc and VTA were delineated based on low-magnification images of TH immunostaining. Stereological counts were made of DA neurons using the Optical Fractionator Probe in Stereo Investigator (MBF Bioscience) at regular predetermined intervals (grid size: x = 170 μm, y = 120 μm) with an unbiased counting frame (x = 55 μm, y = 33.6 μm; dissector height, z = 33.6 μm). The actual mounted section thickness averaged 24 μm (50% shrinkage from the unprocessed section thickness).

## Single-cell reverse transcription PCR

Sampling was done from acute ventral midbrain slices. Anesthetized mice (male or female WT or DAT GLS1 cKO and littermate control mice) were decapitated and brains quickly removed in ice-cold high-glucose artificial cerebrospinal fluid (aCSF; in mM: 75 NaCl, 2.5 KCl, 26 NaHCO$_3$, 1.25 NaH$_2$PO$_4$, 0.7 CaCl$_2$, 2 MgCl$_2$ and 100 glucose, adjusted to pH 7.4). 300 μm coronal midbrain sections were cut on a vibrating microtome (Leica VT1200S). Sections were preincubated for at least one hour at room temperature in high sucrose aCSF saturated with carbogen (95% O$_2$5% CO$_2$), then mounted in a chamber on the stage of an upright microscope (Olympus BX61WI) continuously perfused with standard aCSF (in mM: 125 NaCl, 2.5 KCl, 25 NaHCO$_3$, 1.25 NaH$_2$PO$_4$, 2 CaCl$_2$, 1 MgCl$_2$ and 25 glucose, pH 7.4; perfusion 1 ml/min) saturated with 95% O$_2$5% CO$_2$. Sampling was done from the VTA and SN, using the medial lemniscus as the dividing boundary. Glass pipettes for sampling were fabricated from thin wall glass capillaries (Harvard Apparatus), which were cleaned with water and ethanol and then treated at 200°C for 4 hr to inactivate RNase. Pipettes were filled with 5 μl DEPC treated water. Whole cell recordings were made using digitally enhanced DIC optics, at room temperature (21–23°C). The cytosol of single neurons was aspirated using a glass pipette. In most cases, the nucleus was aspirated along with the cytosol. The sampled single-cell cytosol was ejected in a 0.2 ml PCR tube with a sample mixture of 0.5 μl dithiothreitol (DTT; 0.1 M, Invitrogen), 0.5 μl RNase inhibitor (RNaseOUT, 40 U/ml, Invitrogen), 1 μl random hexamers (50 μM, Applied Biosciences) and 5 μl DEPC treated water. Sampling was done and the tubes with sample mixture were kept on ice until reverse transcription. The sample mixture was treated at 70°C for 10 min. The second mixture (4 μl x5) was added to the sample mixture. First strand buffer (Invitrogen), 0.5 μl RNase inhibitor, 1 μl dNTP mix (10 mM, Invitrogen), 1.5 μl DTT, and 1 μl reverse transcriptase (SuperScript III, 200 U/μl, Invitrogen). Reverse transcription was done at 50°C for 50 min, and stopped by raising the temperature to 85°C for 5 min. Subsequently, 0.5 μl RNase (2 u/μL, Invitrogen) was added to each tube and incubated at 37°C for 20 min to eliminate RNA contamination. The cDNA produced by reverse transcription was frozen at −80°C pending PCR analysis. After reverse transcription, cDNA was amplified by nested PCR. First round PCR primers spanned at least one intron to preclude amplification of genomic DNA. TH and GAD67 primer sequences for both first and second round PCR were obtained from *Liss et al. (1999)*; VGLUT2 primer sequences for the second round

were obtained from *Mendez et al. (2008)*. VGLUT2 first round primers and GLS1 primers for both the first and second round PCR were custom designed, with the following sequences (5' to 3'): VGLUT2 first round upper cacccgcccaaataccacgg and lower gccccaaagacccggttagc; GLS1 first round upper ttgttgtgacttctctaat and lower atggtgtccaaagtgtag; GLS1 second round upper gtggcatgtatgacttct and lower atggtgtccaaagtgtag. Products of the second round PCR were confirmed by sequencing, and had the following sizes (in bp): TH 377, GAD67 702, VGLUT2 250 and GLS1 512. Both first and second round amplifications was done with the following temperature cycle: 3 min at 94°C, 35 cycles of 30 s at 94°C, 1 min at 58°C, 3 min at 72°C, followed by 7 min at 72°C. 2 µl of the first round PCR product was used for the second round. PCR products were separated on 1.5% agarose gels. Only clear bands were counted as positive; runs with unclear bands or bands of incorrect size were discarded.

## RNA extraction and reverse transcription quantitative PCR (RT-qPCR)

We used male and female juvenile (P30) DAT$^{IREScre/+}$ and littermate controls. Mice were anesthetized with ketamine/xylazine. The ventral midbrain and dorsal striatum were dissected and put in tubes with 300 µl Qiazol (Qiagen), a RNase-inhibitor buffer, and rapidly frozen on dry ice. RNA extraction was done using the RNeasy Lipid Mini Kit (Qiagen), according to the manufacturer's instructions, and stored in RNase-free water at −80°C until further processing. RNA concentrations were standardized to 1 µg per 10 µl water using a NanoDrop 1000 Spectrophotometer (ThermoScientific). The 260:280 nm absorbance ratio was measured to assess RNA quality; samples were excluded if the ratio was outside the range 2.0 ± 0.2, or if the RNA concentration was too low. Genomic DNA elimination was performed using RNase-free DNase set (Qiagen). Reverse transcription was carried with the RT$^2$ first-strand kit (Qiagen). Reverse transcription product (cDNA) was diluted to a volume of 1 ml in water. The real time quantitative PCR (RT-qPCR) was performed using an Opticon 2 DNA Engine (Bio-Rad) and microprofiler plates with primers designed by SuperArray Biosciences (Qiagen). The primers were custom designed to recognize cDNA for DAT, D1 and D2 receptors, TH, VMAT2. The cycle threshold (Ct) values were normalized to GAPDH ($\Delta$Ct). Relative copy number was obtained by exponentiation of $\Delta$Ct values (function $2^{-\Delta CT}$) multiplied by 1000.

## Quantitative *Gls1* genotyping

We used male and female adult (P90-150) $\Delta$GLS1 HET mice and littermate controls, or EMX1 GLS1 cHETs and littermate controls. Mice were anesthetized with ketamine+xylazine, decapitated and brains quickly removed to ice-cold saline for dissection. The hippocampus, prefrontal cortex, striatum, thalamus and ventral midbrain, dissected from one hemisphere, were put in 96-well plates and sent to Transnetyx (Cordova, TN) for quantitative genotyping using probe-based quantitative PCR (qPCR). Allelic abundance was obtained from the mean of 4 qPCR determinations (2 runs done in duplicate). The floxGLS1 and WT allele signals were normalized to the one-allele signal from floxGLS1 heterozygous mice.

## Slice electrophysiology

Recordings in the NAc shell were made from 300 µm coronal striatal slices, as described previously (*Chuhma et al., 2011*). Animals were anesthetized with ketamine+xylazine. Brains removed into ice-cold high-glucose aCSF saturated with carbogen (95% $O_2$ 5% $CO_2$). The composition of the high-glucose aCSF was, in mM: 75 NaCl, 2.5 KCl, 26 NaHCO$_3$, 1.25 NaH$_2$PO$_4$, 0.7 CaCl$_2$, 2 MgCl$_2$ and 100 glucose, adjusted to pH 7.4. After 1 hr incubation in high-sucrose aCSF at room temperature to allow slices to recover, slices were placed in a recording chamber with continuous perfusion of standard aCSF equilibrated with carbogen, and maintained at 30–32°C (TC 344B Temperature Controller, Warner Instruments). Expression of ChR2 was confirmed by visualization of EYFP fluorescence in DA neuron axons and varicosities. Whole-cell patch recording followed standard techniques using glass pipettes (5–8 M$\Omega$). For voltage clamp experiments, a cocktail of antagonists was included in the perfusate to isolate AMPA-mediated responses: SR95531 10 µM (GABA$_A$ antagonist), CGP55345 3 µM (GABA$_B$ antagonist), SCH23390 10 µM (D1 antagonist), (-)-sulpiride 10 µM (D2 antagonist), scopolamine 1 µM (muscarinic antagonist) and D-AP5 50 µM (NMDA antagonist) (all from Tocris Bioscience). Patch pipettes were filled with intracellular solution containing (in mM) 140 Cs$^+$-gluconate (voltage clamp recordings) or 140 K$^+$-gluconate (current clamp recordings), 10

HEPES, 0.1 $CaCl_2$, 2 $MgCl_2$, 1 EGTA, 2 ATP-$Na_2$ and 0.1 GTP-$Na_2$ (pH 7.3). The $Na^+$-channel blocker lidocaine N-ethyl bromide (QX-314, 5 mM, Sigma-Aldrich) was added to the intracellular solution in voltage clamp experiments to block active currents. For current clamp experiments, no drugs were added to the perfusate. Recordings were made with an Axopatch 200B (Molecular Devices); for voltage clamp recordings (holding potential −70 mV), series resistance (6–35 MΩ) was compensated online by 70–80%. Liquid junction potentials (12–15 mV) were adjusted online. ChR2 responses were evoked by field illumination with a high-power blue (470 nm) LED (ThorLabs). GLU mediation was confirmed by blockade with 40 µM 6-cyano-7-nitroquinoxaline-2,3-dione (CNQX, Tocris Bioscience). Data were filtered at 5 kHz with a 4-pole Bessel filter, digitized (InstruTECH ITC-18 Interface, HEKA) at 5 kHz, and analyzed using Axograph X (Axograph Scientific).

Recordings from putative DA neurons in adult (P59–P64) DAT GLS1 cHET mice and CTRL littermates were made in 300 µm VTA/$SN_c$ horizontal slices, blinded to genotype. The medial optic tract defined the boundary between the $SN_c$ and the VTA. SNc neurons showing slow pacemaker firing and a prominent $I_h$ were identified as DA neurons; in the lateral VTA, large neurons with slow pacemaker firing and a prominent $I_h$ were always DAT-driven reporter positive (Chuhma, unpublished observation). VTA neurons in the medial VTA with these properties are not always TH+ (*Margolis et al., 2010*), so VTA recordings were restricted to the lateral VTA. Whole-cell patch recordings were made with borosilicate glass pipettes (3–6 MΩ) with intracellular solution containing (in mM): 135 $K^+$-methanesulfonate, 5 KCl, 2 $MgCl_2$, 0.1 $CaCl_2$, 10 HEPES, 1 EGTA, 2 $Na_2$-ATP, 0.1 GTP (pH 7.3), using an Axopatch 200B in fast current clamp mode. Since DA neurons were spontaneously active, resting membrane potential was measured as the average of the pacemaker fluctuation of the membrane potential after action potentials were truncated. Input impedance was measured with −100 pA current pulses. Action potential threshold was determined as the point where membrane potential change exceeded 10 mV/ms, using AxographX automatic detection.

## Fast-scan cyclic voltammetry

Recordings were done in adult (P71–P85) DAT GLS1 cHET::ChR2 and CTRL::ChR2 mice, in 300 µm coronal slices through the striatum, as described previously for the electrophysiology experiments. DA release was evoked by photostimulation (blue high-power LED) and measured using carbon fiber electrodes, calibrated to 1 µM DA, post-experiment. A triangle wave (−450 to +800 mV at 312.5 V/sec vs. Ag/AgCl) was applied to the electrode at 10 Hz. Fibers were conditioned in the brain slice by cycling the fiber for 20–30 min or until the current stabilized. Current was recorded using an Axopatch 200B filtered at 10 kHz with a 4-pole Bessel filter, digitized at 25 kHz (ITC-18) using Igor Pro 6 (WaveMetrics) and analyzed with MATLAB R2014b (MathWorks). The apparent DA oxidation peak in response to the applied triangle wave in cyclic voltammetry is determined by the pipette offset of the amplifier, which is used to correct for resistance between the ground and carbon fiber electrodes. In this study, we made no pipette offset adjustment, which decreased the instances that the current signal overloaded the amplifier. However, the DA oxidation peak, which is normally reported at ~+600 mV with a typical offset of ~+150 mV, was left shifted and occurred at ~+400 mV without an offset. We found in calibrations that the absence of pipette offset did not affect the peak amplitude of the DA response.

## DA and DOPAC content

To measure tissue DA and DOPAC content, mice underwent cervical dislocation; brains were removed rapidly and flash frozen in isopentane. Tissue samples were obtained from 1 mm circular punches from approximately 1 mm thick coronal sections, weighed, placed in 200 µl of HeGA preservative solution (0.1 M Acetic Acid, 0.105% EDTA, 0.12% Glutathione, pH 3.7), homogenized (150 VT Ultrasonic homogenizer; Homogenizers.net), centrifuged and supernatant frozen at −80°C pending analysis. Samples were separated by HPLC coupled to an electrochemical detector. DA and DOPAC were separated with a reverse phase C18 column (ChromSep SS 100 × 3.0 mm, Inertsil 3 ODS-3; Varian, Palo Alto, CA) and a mobile phase containing: 75 mM $NaH_2PO_4$, 25 mM citric acid, 25 µM EDTA, 100 µl/L tetraethylamine, 2.2 mM octanesulfonic acid sodium salt, 10% acetonitrile, 2% methanol, pH 3.5. DA was oxidized with a coulometric electrode (Model 5014; ESA, Chelmsford, MA), with conditioning cell set to a potential of −150 to −200 mV and the analytical cell set to a potential of 350 mV. The concentration of DA and its metabolites was quantified using an external

standard curve from standards prepared in the same aCSF/preservative mixture as the brain dialysates.

## PAG protein determination

Protein analysis was performed using the Simon Simple Western assay (ProteinSimple). Hippocampal tissue samples were dissected and homogenized in 100 μL lysis solution. Lysis solution was prepared by mixing 1 mL of 1x lysis buffer (Cell Signalling Technology, 9803) containing 1 μl calyculin A and 0.5 μl okadaic acid (protein phosphatase inhibitors from Sigma-Aldrich, C5552 and 08010 respectively) and 5 μl of protease inhibitor cocktail (Sigma-Aldrich, P8340). After homogenization, the lysate was centrifuged at 12,000 rpm for 30 min at 4°C. The supernatant was transferred to new tubes and frozen at −80°C pending subsequent analysis. Tissue samples were diluted to a concentration of 0.2 mg/mL in ProteinSimple sample buffer. A master mix containing 10x sample buffer, 1M DTT, and 10x fluorescent standard was added to the samples, which were then loaded in the first row of a ProteinSimple cassette. A mixture of two rabbit polyclonal antibodies was loaded in the second row: PAG antiserum (*Curthoys et al., 1976*) diluted 1:200 and GAPDH (14C10) (Cell Signalling Technology, 2118S; AB Registry ID: AB_2107301) diluted 1:25. The luminol-S/peroxide chemiluminescent detection mixture was loaded in the third row. Size-based separation, immunoprobing, washing, and detection were done automatically by the Simon, which in an automated sequence drew up the sample mixture, the antibodies, and then the detection reagent into a capillary array. Chemiluminescence was measured along the length of the capillary over time, and analyzed using ProteinSimple Compass software.

## Behavior

### Motor performance

A rotarod apparatus (accelerating model; Ugo Basile, Varese, Italy) was used to measure motor learning and coordination. Mice were trained in the accelerating speed mode at 0–20 (Day 1), 0–30 (Day 2), and 0–40 (Day 3) rpm, received three trials per day, and performance was expressed as the time to the first fall.

### Novelty-induced locomotion, amphetamine-induced hyperlocomotion and sensitization

Novelty-induced exploration and reactivity to amphetamine were assessed in the open field (Plexiglas activity chambers, 40.6 cm long ×40.6 cm wide × 38.1 cm high; SmartFrame Open Field System, Kinder Scientific, Poway, CA) equipped with infrared detectors to track animal movement. Testing took place under bright ambient light conditions. Novelty-induced activity was recorded for 60 min, after which mice received an i.p. injection of d-amphetamine hemisulfate (Sigma-Aldrich, A5880) or vehicle (saline) and were returned to the open field for 90 min. This protocol was repeated for the amphetamine sensitization studies.

### Anxiety

Anxiety was measured in an elevated plus maze with two open arms and two closed arms linked by a central platform. Two different size mazes were used, a smaller one with shorter arms (28 cm) and 31 cm above the floor, and a larger one with longer arms (45 cm) and 50 cm above the floor. Mice were put in the center of the maze and allowed to explore for 5 min. Behavior was recorded with a video camera located above the maze. In the smaller maze, the time spent in the open arms was scored using TopScan (CleverSys, Reston, VA). In the larger maze, the time spent in the proximal and distal open arms, and the number of entries into the open arms was scored using AnyMaze (Stoelting, Wood Dale, IL).

### Fear conditioning

Fear conditioning was assessed in rodent test chambers (20 cm length x 16 cm width x 20.5 cm height; Med Associates, Fairfax, VT), equipped with a ceiling and wall light, a speaker and a grid floor through which mild electrical shocks were delivered. FreezeFrame video tracker (Coulbourn Instruments, Holliston, MA) was used to measure freezing during the 3 phases of the procedure: conditioning (Day 1), tone test (Day 2), and context test (Day 3). The same context was used for Day

1 and 3 (lemon scent, grid floor and metal hall exposed, ceiling light on and wall light off), while a different context was use on Day 2 (cinnamon scent, colored plastic sheets covered the floor and halls, ceiling light off and wall light on). On Day 1, mice received 3 pairings of a tone (CS; 20 s, 80 dB) and shock (US; 1 s, 0.5 mA). On Day 2, the tone CS was delivered twice (for 20 s at 120 and 200 s after the start of the session) during a 4 min session in a different context, without the contextual cues associated with the shock US. On Day 3, mice were tested for conditioned fear to the training context during a 4 min session, without the tone CS or shock US. Sessions (4 min) were scored for freezing behavior.

## Latent inhibition

Latent inhibition was assessed in the same test chambers used for fear conditioning. Freezing was monitored during the four phases of the paradigm: pre-exposure and conditioning (Day 1), context test (Day 2) and tone test (Day 3). The preexposure stimulus/conditioned stimulus was an 80 dB tone and the unconditioned stimulus was a 1 s, 0.70 mA shock. Mice were randomly assigned to a non-pre-exposed group (NPE; received 3 CS/US pairings on Day 1) and a pre-exposed group (PE; received 20 CS followed by 3 CS/US pairings on Day 1). On Day 1, 30 min before the behavioral test, mice received clozapine (Sigma-Aldrich C6305, 1.5 mg/kg i.p., dissolved in a mixture of 1.5% DMSO and saline) or vehicle. Clozapine is used as a positive control to demonstrate that the limited number of pre-exposures does not elicit LI and yet are sufficient to reveal potentiation of LI, thereby maximizing the dynamic range of the potentiation. On Day 3, mice were put in a different context to measured freezing to the tone, which was presented for 8 min. The same context was used for Days 1 and 2 (lemon scent, grid floor and metal hall exposed, ceiling light on and wall light off), while a different context was use on Day 3 (anise scent, colored plastic sheets covered the floor and halls, ceiling light off and wall light on). The scent was delivered to the chambers by placing a paper towel dabbed with the scent solution under the chamber floor.

The LI procedure was conducted over three days:

*Day 1: Preexposure/Conditioning* - Preexposed (PE) mice received 20 presentations of a 30 s tone CS at a variable interstimulus interval of 30 s; while the non-preexposed (NPE) mice were confined to the chamber for an identical period of time without receiving the CS. Conditioning began immediately upon completion of the PE in the same chamber, and comprised 3 tone-shock CS-US pairings, given 3 min apart. Each trial began with the 30 s tone CS; a foot shock immediately followed tone termination. Mice were observed for freezing. After the last pairing, mice remained in the chamber for an additional 5 min.

*Day 2: Context Test* - Mice were tested for conditioned fear of the training context. Mice were placed in the experiment chamber for 8 min and presented with neither tone nor shock and observed for freezing.

*Day 3: Tone Test* - Mice were tested for conditioned fear induced by the tone presentation in absence of the contextual cues associated with shock. Each mouse was placed in the chamber for 12 min. After an acclimatization period of 3 min, the tone CS was delivered for 8 min (no shocks were administered), and mice observed for freezing.

## Sample size estimation

Sample size estimates were made using G*Power (*Faul et al., 2007*). Sample sizes were calculated using a power of 0.80 and an $\alpha$ of 0.05, as we assumed that a 4:1 ratio between type 1 and type 2 errors was appropriate for all our experiments (*Keppel, 1991* p. 75). The predicted effects sizes were different for the behavioral, electrophysiology/voltammetry, and stereology experiments. Since we were assessing the effects of a conditional heterozygous manipulation, for the behavioral studies we predicted a medium effect size of 0.06 (partial $\eta^2$), which resulted in an estimated sample size range between 17 to 51 mice per group (rotarod = 17; elevated plus maze short arms = 51; novelty-induced locomotion = 22; amphetamine sensitization = 21). After running these first experiments in sequence using samples sizes within the estimated range, we obtained significant F values with effects sizes ranging from 0.06 to 0.15 and a better than predicted power of 0.9, which led us to use smaller samples size in subsequent experiments (elevated plus maze longer arms, acute amphetamine, fear conditioning, latent inhibition). For the electrophysiology and voltammetry studies, which measured the direct effects of the conditional heterozygous manipulation on synaptic release, we

predicted an effect size of 0.1, which resulted in an estimated sample size of 12 per group. For the stereology experiments, we estimated a larger effect size of 0.2 based on previous experiments and pilot studies, for a sample size of 4 per group.

## Statistical analysis

In *Figure 1*, the stereological estimate of the number of TH$^+$ only, PAG$^+$ only and TH$^+$ / PAG$^+$ cells in the VTA and SNc of juvenile wild-type mice was analyzed using a 3 (cell type) X 2 (brain region) ANOVA. For the comparison between the relative number of TH$^+$ / PAG$^+$ cells in juvenile (P25) and adult (P60) mice, a 2 (age) X 2 (brain region) ANOVA was used. For the single cell RT-PCR data, the Chi-Square test was used to determine whether TH$^+$ / VGLUT2$^+$ neurons preferentially expressed PAG.

In *Figure 3*, comparison of response amplitudes to single photostimulation was analyzed using a 2 (genotype) X 2 (cell type) ANOVA. Comparison between genotypes of first response amplitude to burst photostimulation was done for each cell type separately using the nonparametric Mann-Whitney test, since samples were not normally distributed. For the analysis of the amplitude of EPSCs induced by repeated burst photostimulation, data was converted to percent of the first response amplitude and analyzed for each cell type separately using a 2 (genotype) X 4 (pulses, repeated measures factor) ANOVA. Only the results obtained from ChIs revealed a significant genotype X pulses interaction, which led us to conduct further analysis of simple effects involving the non-repeated measures factor (genotype) to detect the source of the interaction. To control for increased family-wise type 2 errors due to multiple comparisons, we applied the Bonferroni correction for simple effects and using $\alpha = 0.0125$. Finally, the analysis of the ratio of firing during burst (0–0.5 s from onset of train) and just after burst photostimulation (0.5–1 s from onset) was done using a one-way ANOVA.

In *Figure 4*, genotypic differences in numbers of TH$^+$ neurons, DA content and DOPAC/DA ratio values were evaluated with one-way ANOVAs. For voltammetry data, the peak amplitude of DA release evoked by consecutive bursts of photostimulation followed by a single, or consecutive single pulses followed by burst, was analyzed using a 2 (genotype) x 4 (pulses, repeated measures factor) ANOVA.

In *Figure 5*, the latency to fall from the rotarod was analyzed using a 2 (genotype) x 9 (trials, repeated measures factor) ANOVA. Locomotor counts in the open field were analyzed using a 2 (genotype) x 6 (time, bins of 10 mins, repeated measures factor) ANOVA. Behavior in the elevated plus maze and fear conditioning chambers was analyzed using a one-way ANOVA to evaluate genotypic effects. Dose effects in amphetamine-induced locomotion were analyzed using a 2 (genotype) x 3 (dose) ANOVA.

In *Figure 6*, for the sensitization experiment, locomotor activity during the first 2 habituation days (vehicle injections) was analyzed separately using a 2 (genotype) x 2 (drug treatment) x 2 (days, repeated measure factor) ANOVA. Locomotor activity during the subsequent 5 test days (vehicle or amphetamine injections) was analyzed using a 2 (genotype) x 2 (drug treatment) x 5 (days, repeated measure factor) ANOVA. A significant three-way interaction was further analyzed for simple effects. Within each drug treatment, a 2 (genotype) x 5 (days, repeated measure factor) ANOVA was used. Only within the amphetamine-treated groups was there a significant genotype X day interaction, which allowed us to conduct a further analysis of simple effects involving the non-repeated measures factor (genotype). Comparisons during the last 3 days of injections were corrected by a Bonferroni adjustment ($\alpha = 0.016$). The data from the challenge day were analyzed separately using a 2 (genotype) X 2 (drug treatment) ANOVA. In addition, data obtained during the 90 min following injections was analyzed separately for each amphetamine- and vehicle-treated group using a 2 (genotype) x 9 (time, bins of 10 min, repeated measure factor) ANOVA.

For the latent inhibition experiment (in *Figure 6*), freezing before CS presentation was analyzed separately for each genotype using a 2 (preexposure treatment) X 2 (time, bins of 1 min, repeated measure factor) ANOVA. After CS presentation data were analyzed using a 2 (preexposure treatment) X 8 (time, bins of 1 min, repeated measure factor) ANOVA. A significant preexposure X time interaction was found for DAT GLS1 cHET mice, allowing us to examine simple effects. The multiple comparisons for each 1 min time bin after CS presentation were corrected by a Bonferroni adjustment ($\alpha = 0.006$). The data for total amount of freezing during the CS were analyzed using a 2 (genotype) X 2 (preexposure treatment) ANOVA. We found a significant genotype X preexposure

interaction, allowing us to explore further the source of the interaction within each genotype using one-way ANOVAs.

A few mice were removed from experiments because of procedural errors (mice were put in the wrong treatment group, or tested in the wrong operant box).

## Acknowledgements

We thank Shannon Wolfman, Celia Gellman, Benjamin Inbar, Lauren Rosko, Karin Krueger, Leora Boussi and Sophia Tepler for technical assistance, Eugene Mosharov, Se Joon Choi, Hadassah Tamir, Benjamin Klein and David Hirschberg for advice, Norman Curthoys for glutaminase antisera, and Theresa Swayne in The Confocal and Specialized Microscopy Shared Resource of the Herbert Irving Comprehensive Cancer Center at Columbia University, supported by NIH grant P30 CA013696. This work was supported by a NARSAD Young Investigator award (SM), DA017978 and MH087758 (SR) and MH086404 (SR, HM).

## Additional information

### Funding

| Funder | Grant reference number | Author |
|---|---|---|
| National Institute on Drug Abuse | MH087758 | Stephen Rayport |
| National Institute on Drug Abuse | DA017978 | Stephen Rayport |
| National Alliance for Research on Schizophrenia and Depression | Young Investigator Award | Susana Mingote |
| National Institute of Mental Health | MH086404 | Holly Moore Stephen Rayport |

The funders had no role in study design, data collection and interpretation, or the decision to submit the work for publication.

### Author contributions

SM, Conceptualization, Data curation, Formal analysis, Funding acquisition, Investigation, Methodology, Writing—original draft, Writing—review and editing; NC, Data curation, Investigation, Writing—review and editing; AK, Data curation, Investigation; GMT, YW, AM, CS, IZ-S, A-CS, Investigation; MGW, Resources, Methodology; JL-O, Resources; DS, Resources, Supervision, Methodology; HM, Supervision, Funding acquisition, Methodology; IG-S, Conceptualization, Supervision, Methodology; SR, Conceptualization, Data curation, Formal analysis, Supervision, Funding acquisition, Project administration, Writing—review and editing

### Author ORCIDs

Susana Mingote, http://orcid.org/0000-0002-0401-4317
Caroline Sferrazza, http://orcid.org/0000-0001-5861-111X
David Sulzer, http://orcid.org/0000-0001-7632-0439
Stephen Rayport, http://orcid.org/0000-0001-9755-7486

### Ethics

Animal experimentation: This study was performed in strict accordance with the recommendations in the Guide for the Care and Use of Laboratory Animals of the National Institutes of Health, under protocols approved by the Institutional Animal Care and Use Committees of Columbia University (# AC-AAAB2862) and New York State Psychiatric Institute (# 1249). All surgery was performed under ketamine + xylazine anesthesia, and every effort was made to minimize suffering.

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
