## [Decision Letter]

Thank you for submitting your article "Dopamine neuron dependent behaviors mediated by glutamate cotransmission" for consideration by *eLife*. Your article has been favorably evaluated by Eve Marder (Senior Editor) and three reviewers, one of whom, Sacha B Nelson, is a member of our Board of Reviewing Editors. The following individual involved in review of your submission has agreed to reveal their identity: Veronica Alvarez (Reviewer).

The reviewers have discussed the reviews with one another and the Reviewing Editor has drafted this decision to help you prepare a revised submission.

Summary:

The authors address the role of glutamatergic cotransmission in dopamine (DA) neurons using an elegant genetic approach that minimizes developmental effects and compensation and primarily affects transmission at high rates (DATcre driven deletion of the glutamate recycling enzyme Gls1). This manipulation does not affect DA neuron function or DA neuron dependent behaviors in multiple assays, but does affect amphetamine sensitization and latent inhibition; two behavioral phenotypes linked to human psychosis and animal models of schizophrenia. Strikingly, deletion of this enzyme in forebrain glutamatergic neurons does not produce either phenotype. Although a prior paper from this group showed some of the behavioral effects demonstrated here in a global manipulation of Gls1, the present results are especially significant as they provide a potential pathophysiological unification of dopaminergic and glutamatergic theories of schizophrenia.

Essential revisions:

1) The reviewers agreed that the paper would be improved if the impact of the genetic manipulation on co-transmission were assessed at an intermediate frequency (between 0.1 and 20 Hz) that more closely resembles the tonic firing of DA neurons in vivo (e.g. 2-5 Hz). However, after discussion, the reviewers agreed that this point could potentially be handled by softening the conclusion that it is burst and not tonic firing that is responsible and increasing discussion of this point. However, it was felt that including these data would be strengthen the paper even if the result were somewhat intermediate between effects at 0.1 and 20 Hz (especially given the vagaries of relating frequencies of optogenetic stimulation in vitro to firing frequencies in vivo).

2) One of the reviewers was concerned that subtle changes in DA release, perhaps occurring only during sensitization, would not have been detected and could contribute to the observed behavioral results. Further discussion between the reviewers suggested that effectively ruling out this possibility experimentally would be difficult, but that it should be discussed.

3) The reviewers raised a number of additional concerns and desires for clarification which are not strictly "minor" but which should require only textual changes in the manuscript. These are listed below.

Additional points:

Reviewer #1:

The Discussion would benefit from a few additional sentences about the significance of the findings for theories of the pathophysiology of schizophrenia aimed at the general reader, particularly as they result to prior studies on Gls1 loss of function and to the relationship between theories emphasizing the roles of dopaminergic and glutamatergic systems in schizophrenia.

"schizophrenia resilience phenotype" is a rather specialized term that should be explained for the general reader earlier in the manuscript.

"cHET" should be defined the first time (or two) it is used – even though HET is clearly defined.

Reviewer #2:

1) Is the pause in Chl firing reduced in duration in DAT GLS1 HET mice? Can this be quantified? It seems like it in the example and I wonder whether this is a consequence of less excitation or less DA mediated inhibition. The study later shows no difference in evoked DA transients (Figure 4) which might say it is not due to changes in DA mediated signaling but it could be confirmed if there is reduction in the duration.

2) Figure 4. The example voltammogram shows that the oxidation current peaks at 400 mV, however, the dopamine oxidation peak is near 600 mV. Is this an issue of electrode offset?

3) Figure 6. In the amphetamine sensitization figure, the data on the vehicle day before the challenge is mentioned in the text but it is not shown in the figure. It is the only data missing and it would be important to display it in Figure 6 to emphasize that the contextual conditioned response. On that note, the locomotor counts double in the control mice so it seems unfair to describe the context response as "very small".

4) On that same topic of the context conditioned response, the stats show no interaction or effect of genotype but the response to the amphetamine paired context seems also smaller in the HET in proportion to the vehicle treated group of each genotype. I wonder how the comparison would result if performed like in 6C but with a graph for each genotype comparing the locomotor response in the vehicle day for vehicle- and amphetamine-pretreated mice. If not significant different for the HET mice, a poor context-conditioned response can account in part for the difference seen in the amphetamine challenge day.

5) It appears like the effect of clozapine on the latent inhibition is occluded in DAT-GLS1 HET mice, or max-up as indicated in the results (Figure 6—figure supplement 5). I would suggest using the term occlusion and also reviewing that sentence because there is an extra misplaced word there. These findings are to the possible mechanism underlying the response of DAt-GLS1 HET mice so I would suggest moving this to the main figure. Also, please add label somewhere to make clear these data are after clozapine injection and it can be helpful to use different symbols to mark the statistical difference or lack of between the panel A and B.

Reviewer #3:

The interpretation that glutamate released during burst firing contributes to amphetamine sensitization is somewhat inconsistent with past work. Several past studies have shown that knockout of NMDA receptors leads to a dramatic reduction in burst firing, but has no effect on cocaine sensitization.

Second, a very recent study tested intracranial self-stimulation in cKOs of VGluT2 in dopamine neurons over many days/sessions and found only very small reduction in self-stimulation enforced place preference (Wang et al., 2017). Please discuss this.

---

## [Author Response]

Essential revisions:

1) The reviewers agreed that the paper would be improved if the impact of the genetic manipulation on co-transmission were assessed at an intermediate frequency (between 0.1 and 20 Hz) that more closely resembles the tonic firing of DA neurons in vivo (e.g. 2-5 Hz). However, after discussion, the reviewers agreed that this point could potentially be handled by softening the conclusion that it is burst and not tonic firing that is responsible and increasing discussion of this point. However, it was felt that including these data would be strengthen the paper even if the result were somewhat intermediate between effects at 0.1 and 20 Hz (especially given the vagaries of relating frequencies of optogenetic stimulation in vitro to firing frequencies in vivo).

We thank the reviewers for making this important point. We agree that examining the effects of an intermediate stimulation frequency to support the conclusion of a selective effect on phasic, and not on tonic, firing would be informative. However, this would introduce a considerable delay, as it would require breeding a new cohort of mice (minimum 10 of each genotype) and then comparing directly 5 Hz stimulation with 20 Hz stimulation. The comparison between 0.1 and 20 Hz made in the current manuscript, highlights the frequency dependence of GLS1 effects, in agreement with previous reports that PAG plays a crucial role in recycling glutamate during sustained activity (Billups et al., 2013; Gaisler-Salomon et al., 2009; Masson et al., 2006; Tani et al., 2014). For this reason, we have modified the manuscript to soften the argument that the effect was selective to phasic firing and emphasize the frequency-dependent effect of the PAG reduction. We have taken “phasic” out of the title. These alterations do not affect the main conclusion or implications of our findings regarding the function of dopamine neuron GLU cotransmission and schizophrenia, but do leave the determination of the specific role of phasic GLU cotransmission for future study.

2) One of the reviewers was concerned that subtle changes in DA release, perhaps occurring only during sensitization, would not have been detected and could contribute to the observed behavioral results. Further discussion between the reviewers suggested that effectively ruling out this possibility experimentally would be difficult, but that it should be discussed.

We appreciate this important point. Our data indicate that DA neuron DA signaling is unaffected in DAT GLS1 cHETS, at baseline, prior to amphetamine administration. Changes in DA signaling are likely to parallel the more modest sensitization seen in the DAT GLS1 cHETS, and in future studies this would be very interesting to examine. To make this point clearer, we have included the following Discussion:

“DA signaling increases with psychostimulant sensitization {Vezina, 2004, 827-39; Bocklisch, 2013, 1521-1525; Covey, 2014, 200-210}. […] While subsequent circuit effects impacted by reduced GLU cotransmission involve DA signaling, we show here for the first time that the attenuation of GLU cotransmission in the absence of developmental alterations and direct effects on DA transmission has strong and selective behavioral effects, revealing a new mechanism through which DA neurons control behavior.”

3) The reviewers raised a number of additional concerns and desires for clarification which are not strictly "minor" but which should require only textual changes in the manuscript. These are listed below.

Additional points:

Reviewer #1:

The Discussion would benefit from a few additional sentences about the significance of the findings for theories of the pathophysiology of schizophrenia aimed at the general reader, particularly as they result to prior studies on Gls1 loss of function and to the relationship between theories emphasizing the roles of dopaminergic and glutamatergic systems in schizophrenia.

We have extended the Discussion on the significance of our findings for the pathophysiology of schizophrenia:

“In the context of schizophrenia, the behaviors affected in DAT GLS1 cHETs align with the schizophrenia resilience phenotype of stopGLS1 HET {Gaisler-Salomon, 2009, 2305-2322}, as well as ΔGLS1 HET mice, both with a global GLS1 heterozygous reduction. […] This inconsistency would be reconciled if increased activity in the VTA and NAc in SCZ is associated with a pathological increase in GLU cotransmission with less of an increase in DA transmission.”.

"schizophrenia resilience phenotype" is a rather specialized term that should be explained for the general reader earlier in the manuscript.

We added the following to the Introduction:

“The global heterozygous *GLS1* reduction impacts several DA dependent behaviors that underpin a schizophrenia resilience phenotype {Gaisler-Salomon, 2009, 2305-2322}, characterized by an attenuated response to psychostimulant challenge, potentiated latent inhibition, procognitive effects {Hazan, 2014, 1916-1924}, together with CA1 hippocampal hypoactivity inverse to the CA1 hyperactivity seen in patients with schizophrenia {Schobel, 2009, 938-946; Gaisler-Salomon, 2009, 1037-1044}. Genetic mutations engendering resilience carry strong therapeutic valence as they directly identify therapeutic targets {Mihali, 2012, 785-799}.”

"cHET" should be defined the first time (or two) it is used – even though HET is clearly defined.

cHET is now clearly defined in the Introduction:

“Here we show in DAT GLS1 conditional heterozygous (cHET) mice – with a DAT-driven- GLS1 reduction – that DA neuron GLU cotransmission is reduced in a frequency dependent manner, without affecting DA neuron development or DA release, and that behaviors that rely on the motivational salience-encoding function of DA neurons are selectively affected, with implications of DA neuron GLU cotransmission for schizophrenia pharmacotherapy.”

Reviewer #2:

1) Is the pause in Chl firing reduced in duration in DAT GLS1 HET mice? Can this be quantified? It seems like it in the example and I wonder whether this is a consequence of less excitation or less DA mediated inhibition. The study later shows no difference in evoked DA transients (Figure 4) which might say it is not due to changes in DA mediated signaling but it could be confirmed if there is reduction in the duration.

In agreement with our previous report (Chuhma et al., 2014), we found that train photostimulation of DA terminals induces burst firing of ChIs followed by a reduction or pause in firing. To show the effect of the GLS1 reduction on the post-burst reduction in firing frequency, we added a color-table to Figure 3 showing all the responses, confirming the lack of genotypic effect during the post-burst interval. The updated Results now read:

“In the subsequent half-second window, the firing ratio reversed to below baseline in CTRL (0.6 ± 0.08) and cHETs (0.7 ± 0.11), which did not differ (Figure 3, right). […] Thus, PAG plays an important role in sustaining DA neuron GLU cotransmission at higher firing frequencies and determines their ability to drive Chls to fire in bursts.”

2) Figure 4. The example voltammogram shows that the oxidation current peaks at 400 mV, however, the dopamine oxidation peak is near 600 mV. Is this an issue of electrode offset?

Yes, and we added a sentence to the methods FSCV section to clarify this issue:

"The apparent DA oxidation peak in response to the applied triangle wave in cyclic voltammetry is determined by the pipette offset of the amplifier, which is used to correct for resistance between the ground and carbon fiber electrodes. In this study, we made no pipette offset adjustment, which decreased the instances that the current signal overloaded the amplifier. However, the DA oxidation peak, which is normally reported at ~+600 mV with a typical offset of ~+150 mV, was left shifted and occurred at ~+400 mV without an offset. We found in calibrations that the absence of pipette offset did not affect the peak amplitude of the DA response."

3) Figure 6. In the amphetamine sensitization figure, the data on the vehicle day before the challenge is mentioned in the text but it is not shown in the figure. It is the only data missing and it would be important to display it in Figure 6 to emphasize that the contextual conditioned response. On that note, the locomotor counts double in the control mice so it seems unfair to describe the context response as "very small".

When we mentioned that it was a “very small” response, this was in comparison to the amphetamine challenge response. This effect is probably due to the fact that drug-induced conditioned responses tend to be smaller when mice are tested in a familiar context (Crombag et al., 2001). Mice were exposed to the open field for 2 days (3 hours per day) for the two initial vehicle injections prior to the start of the amphetamine injections, so the novelty of the open field was dramatically reduced and consequently the strength of the conditioned response. To show this data, we have added the vehicle challenge data to Figure 6, both for the DAT GLS1 cHET mice and EMX1 GLS1 cHET mice, and changed the Results to include:

“The vehicle challenge revealed a modest but significant conditioned response in the Amph-treated groups.”

The statistical analysis is now reported in the Figure 6 caption.

4) On that same topic of the context conditioned response, the stats show no interaction or effect of genotype but the response to the amphetamine paired context seems also smaller in the HET in proportion to the vehicle treated group of each genotype. I wonder how the comparison would result if performed like in 6C but with a graph for each genotype comparing the locomotor response in the vehicle day for vehicle- and amphetamine-pretreated mice. If not significant different for the HET mice, a poor context-conditioned response can account in part for the difference seen in the amphetamine challenge day.

We analyzed the vehicle challenge day as was done for Figure 6. A repeated measures ANOVA for the Veh-treated mice show no main effect of genotype (F_(1,34)_=0.454, p=0.51) or significant interaction (F_(4.5,320)_=1.578, p=0.176), but a main effect of time (F_(4.5,320)_=13.358, p<0.001). For the Amph-treated mice, we found similar results, no main effect of genotype (F_(1,40)_=0.052, p=0.82) or significant interaction (F_(3.2,320)_=2.058, p=0.103), but a main effect of time (F_(3.2,320)_=34.44, p<0.001). So, cHET did develop a conditioned response after repeated Amph treatment to the same extent as seen in CTRL mice.

5) It appears like the effect of clozapine on the latent inhibition is occluded in DAT-GLS1 HET mice, or max-up as indicated in the results (Figure 6—figure supplement 5). I would suggest using the term occlusion and also reviewing that sentence because there is an extra misplaced word there. These findings are to the possible mechanism underlying the response of DAt-GLS1 HET mice so I would suggest moving this to the main figure. Also, please add label somewhere to make clear these data are after clozapine injection and it can be helpful to use different symbols to mark the statistical difference or lack of between the panel A and B.

We have changed the text in the Results regarding Figure 6—figure supplement 5:

“To confirm in EMX1 GLS1 cHETs that the limited number of pre-exposures did not elicit LI and yet was sufficient to reveal potentiation of LI, we tested for clozapine-induced potentiation of LI {Gaisler-Salomon, 2009, 2305-2322} (Figure 6—figure supplement 5). […] The lack of further potentiation of LI in ΔGLS1 HET and DAT GLS1 cHET mice suggests that clozapine treatment and GLS1 deficiency in DA neurons each either induce maximal potentiation of LI, or involve shared mechanisms so that GLS1 deficiency occludes clozapine-induced potentiation of LI.”

We also added a schematic of the latent inhibition protocol in Figure 6—figure supplement 5 to show the time of administration of the single clozapine dose on Day 1 before the start of the procedure. We prefer to keep this figure as a supplement since the clozapine experiment was done as a positive control to show that EMX1 GLS1 cHET mice given the same number of tone preexposues as the other HET genotypes would show LI potentiation. For both the ΔGLS1 HET and DAT GLS1 cHET mice, the GLS1 deficiency occluded the clozapine effect. It would be challenging to explore the implications of this result further since clozapine acts on multiple receptors and several mechanism could be responsible for the absence of a pharmacological effect. Moreover, the simplest explanation is that there was a floor effect on LI potentiation.

Reviewer #3:

The interpretation that glutamate released during burst firing contributes to amphetamine sensitization is somewhat inconsistent with past work. Several past studies have shown that knockout of NMDA receptors leads to a dramatic reduction in burst firing, but has no effect on cocaine sensitization.

In relation to the NR1 cKO study of Zweifel et al. (2008), it is interesting that they also reported a reduction in the expression of sensitization very similar to what we observed in DAT GLS1 cHET mice. We now discuss this paper in the Discussion:

“DAT GLS1 cHET mice do not show sensitization to amphetamine with repeated administration, and after a withdrawal period show reduced expression of sensitization. […] In contrast, DAT NR1 cKO showed a reduction in conditioned responses to context not seen in the present study, suggesting that the abrogation of both DA and GLU phasic transmission must be affected to impact the development of drug-induced conditioned responses.”

Second, a very recent study tested intracranial self-stimulation in cKOs of VGluT2 in dopamine neurons over many days/sessions and found only very small reduction in self-stimulation enforced place preference (Wang et al., 2017). Please discuss this.

The Wang et al. (2017) results do not contradict our findings, since we do not think that GLU cotransmission drives reinforcement learning. In relation to this article we have added the following to the Discussion:

“The temporal precision of the DA neuron GLU signal makes it particularly suitable for updating salience. […] Instead, our behavioral results suggest that DA neuron GLU signals modulate learning by regulating the attribution of motivational salience to relevant events via their direct control over ChI activity.”